# How Now Oblong Cow:
# Benchmark for Intrinsic Dimension Estimation Using Homogeneous Spaces

## Abstract

Machine learning models can generalize well on real-world datasets. According to the manifold hypothesis, this is possible because datasets lie on a latent manifold with small intrinsic dimension (ID). There exist many methods for ID estimation (IDE), but their estimates vary substantially. This warrants benchmarking IDE methods on manifolds that are more complex than those in existing benchmarks. We propose a Quantum-Inspired Intrinsic-dimension Estimation (QuIIEst) benchmark consisting of infinite families of topologically non-trivial manifolds with known ID. Our benchmark stems from a quantum-optical method of embedding arbitrary homogeneous spaces while allowing for curvature modification and additive noise. The IDE methods tested were generally less accurate on QuIIEst manifolds than on existing benchmarks under identical resource allocation. We observe minimal performance degradation with increasingly anisotropic and non-uniform curvature, underscoring the benchmark's inherent difficulty as well as the importance of anisotropic curvature to benchmarking. As a result of independent interest, we perform IDE on the fractal Hofstadter's butterfly and identify which methods are capable of extracting the effective dimension of a space that is not a manifold.

## 1 Introduction

The success of machine learning (ML) algorithms on datasets with large representative dimensions is often attributed to the hypothesis that the data actually lies on a manifold with smaller dimension, but embedded in a larger space Gorban & Tyukin (2018b); Fefferman et al. (2013); Olah (2014); Cayton (2005). An ML algorithm is able to generalize because it can infer this manifold (or sub-manifold) from the given training distribution. This notion is formalized by the concept of *intrinsic dimension* (ID).

The manifold hypothesis is quite intriguing and has been subject to many experimental investigations, whereby different methods have been proposed to estimate the ID of a particular data-cloud, which have been then applied to popular ML datasets such as MNIST LeCun et al. (1998); Deng (2012). However, a common theme is the disagreement in the estimated IDs of different methods on these real-world datasets Hein & Audibert (2005); Facco et al. (2017); Pope et al. (2021); Bahadur & Paffenroth (2019); Goldt et al. (2020); Candelori et al. (2025); Stanczuk et al. (2023); Tempczyk et al. (2021); Kamkari et al. (2024). This large variability between different methods suggests that either (i) the manifold hypothesis is incorrect, or (ii) these different methods have their inherent biases which may or may not be relevant to particular datasets.

Hence, it is important to benchmark existing methods against datasets consisting of more complicated manifolds whose ground-truth features and IDs are known. There have been a few proposed benchmarks Hein & Audibert (2005); Kamkari et al. (2024); Camastra (2003); Bac et al. (2021); Johnsson (2016) that include manifolds like spheres, hyper-cubes, Swiss rolls, Möbius strips, among others. However, each of these datasets comes with their own flaws — low dimensionality, presence of singular points, and, most importantly, lack of tractable yet non-trivial infinite families of manifolds with varying ID.

| Property | Spheres | Gaussian vectors | Möbius strips | Nonlinear maps on $\mathbb{R}^d$ | Affine spaces | QuIIEst |
|---|---|---|---|---|---|---|
| Anisotropic curvature | ✗ | ✗ | ✗ | ✓ | ✗ | ✓ |
| Scalable manifold families | ✓ | ✓ | ✗ | ✓ | ✓ | ✓ |
| Multiple natural embeddings | ✗ | ✗ | ✗ | ✗ | ✗ | ✓ |
| Non-trivial topology | ✗ | ✗ | ✓ | ✗ | ✗ | ✓ |

Figure 1: While most methods perform well when it comes to intrinsic dimension estimation (IDE) for simplistic manifolds like (hyper-)spheres, there's wide variability in their estimates for real-world datasets like MNIST. We propose QuIIEst- a family of topologically non-trivial manifolds to serve as an **intermediate confidence evaluation** for IDE. The QuIIEst dataset contains several different embeddings of infinite families of manifolds whose dimension is polynomial in their parameters, and which admit nontrivial geometry and topology.

In this light, we propose the **QuIIEst** (**Qu**antum-**I**nspired **I**D **Est**imation) benchmark — a collection of synthetic datasets sampled from non-trivial manifolds constructed with tools from quantum information theory. We highlight our contributions in this regard below.

**Summary of Contributions:**

1. We provide explicit *families of manifold embeddings* with known ground-truth IDs to serve as a benchmark for current and future IDE techniques. Our manifolds sport several advantages over other benchmarks, summarized in Figure 1.
2. Under identical resource allocation, our manifolds are more challenging relative to standard benchmarks, even at low dimensionality. We attribute this to the direction dependent curvature of our manifolds (vis a vis the sphere, whose curvature is direction-*in*dependent).
3. Corroborating the previous point, we observe that IDE performance degrades significantly when we distort the sphere's uniform curvature, and only minimally when we distort our manifolds. This indicates that QuIIEst is *already challenging enough*, as no further distortion is needed.
4. We conjecture that direction-dependent (i.e., anisotropic) curvature is a necessary ingredient for manifolds to resemble real data. To corroborate this, we apply our distortion transformations to MNIST and see that IDE performance degrades only minimally, just like for distorted versions of our manifolds.
5. We provide group-theoretic recipes to embed any manifold that is also homogeneous space, and to further obtain double-coset spaces from such manifolds.

## 2 BACKGROUND AND RELATED WORKS

**Manifolds in ML** The manifold hypothesis states that real-world data lie on low-dimensional manifold, even though represented in high dimensions. Gorban & Tyukin (2018a); Cayton (2005). This idea has been explored both theoretically and experimentally Fefferman et al. (2013); Goldt et al. (2020). Topological properties of such data manifolds Carlsson et al. (2008) and latent representations Ansuini et al. (2019) have also been studied.

**Intrinsic Dimension**  Intrinsic dimension (ID), as the name suggests, characterizes several inherent properties of the manifold Narayanan & Niyogi (2009); Pope et al. (2021). A plethora of estimators have been proposed for ID estimation (IDE), which we discuss in Appx. C. See Campadelli et al. (2015) for a detailed survey.

**Benchmarks for ID Estimation**  Practical benchmarks for IDE involve topologically simple manifolds like hyperspheres or non-scalable manifolds like the Möbius strip Camastra (2003); Hein & Audibert (2005); Rozza et al. (2012); Campadelli et al. (2015); Bac et al. (2021). An interesting benchmark proposed GAN-based data to lower bound ID Pope et al. (2021). Physics-inspired datasets Weigend & Gershenfeld (1993); Camastra & Filippone (2009); Grassberger & Procaccia (1983) typically use non-linear dynamical models. However, the exact ground truth ID is unknown. QuIIEst, as we discuss, alleviates these concerns.

**Quantum and ML**  Unlike quantum machine learning or quantum-inspired algorithms Cerezo et al. (2022); Tang (2019), we take inspiration from quantum optics and quantum information theory to generate diverse, well-defined manifolds for testing ID estimators. Specifically, our benchmark uses Gilmore-Perelomov coherent states Perelomov (1977); Zhang et al. (1990) that correspond to the most "classical" states of a quantum system.

A more comprehensive discussion of previous work may be found in Appx. C.

## 3  PRELIMINARIES

**Manifolds and Intrinsic Dimension**  A topological manifold $M$ of dimension $d_i$ is a topological space that locally "looks" like $\mathbb{R}^{d_i}$, in the sense that there exist local patches that are homeomorphic to $\mathbb{R}^{d_i}$ Lee (2012). For a given disjoint union of manifolds and a point $p$ in this union, the local intrinsic dimension around $p$ is the dimension of the submanifold to which it belongs Kamkari et al. (2024).

In addition to this *intrinsic* viewpoint of a manifold, one can equivalently consider manifolds *extrinsically* by defining them as appropriate subsets of some ambient Euclidean space $\mathbb{R}^{d_a}$ Lee (2012). Given (samples from) a set $S \subset \mathbb{R}^{d_a}$ along with the promise that $S$ is a manifold of dimension $d_i \leq d_a$ for some unknown ID $d_i$, one many naturally desire an algorithmic procedure to estimate $d_i$. Intuitively, the ID of a dataset can be thought of as the minimum number of parameters needed to represent the data with no loss of information Ceruti et al. (2014). Importantly, though, a $d_i$-dimensional manifold can have nontrivial geometry (such as a direction-dependent, or anisotropic, curvature) and topology (such as being not simply connected) that makes it starkly different from real space or a sphere of the same dimension.

Our manifolds, by definition, exhibit a single ID at all points. In contrast, most methods return the local (scale-dependent) ID (LID) estimates at different points. Thus, we will refer to the *mean* of these LID estimates as *the* ID of the manifold. We report the result of experiments with other statistical quantities, such as median and mode, in Appx. G.7.

**Homogeneous spaces**  All manifolds included in the QuIIEst benchmark are parameterized by quotient spaces $\mathcal{G}/\mathcal{H}$ (a.k.a. homogeneous spaces), where $\mathcal{G}$ is a Lie group (i.e., a group that can also be considered as a manifold), and where $\mathcal{H}$ is one of its subgroups. For $\mathcal{G}$, we use either the orthogonal group $O(n)$ or the unitary group $U(n)$, which is the symmetry group of the $n$-dimensional real or complex sphere, respectively. We provide a pedagogical flavor of these spaces below, leaving technical details to Appx. D.

The two-dimensional real sphere, $S^2$, is a simple example of a homogeneous space. There exists a proper (i.e., orientation-preserving) rotation that can map the north pole to any other point on the sphere (whose action can be thought of as moving along the great circle connecting the north pole to the desired point). Therefore, we can label all points on the sphere by the rotations that take us there from the north pole, but we need to omit all rotations that rotate around the north pole since those do not take us anywhere.

Mathematically, this translates to the homogeneous space $SO(3)/SO(2) \cong S^2$, where $SO(3)$ is the group of all proper three-dimensional rotations, and $SO(2)$ is the subgroup of rotations around the north pole.

The characteristics of the quotient space depend on the subgroup, which can be continuous or discrete. Spheres and more general homogeneous spaces whose $\mathcal{H}$ is a continuous group are constructed in the same spirit as this example. Their intrinsic dimension is $d_i = \dim \mathcal{G} - \dim \mathcal{H}$.

The remainder function, for which $\mathcal{G} = \mathbb{R}$ and $\mathcal{H} = \mathbb{Z}$, is an example of a quotient by a discrete, or finite, subgroup. The remainder is obtained from a real number $r$ by subtracting the closest integer less than or equal to $r$, and all remainders lie in the interval $[0, 1)$. This domain is periodic — the remainder cycles as $r$ increases past an integer — demonstrating that $\mathbb{R}/\mathbb{Z} \cong S^1$, the circle.

Homogeneous spaces with finite $\mathcal{H}$ can be thought of as subsets of points of the numerator with higher dimensional periodic identifications. Their intrinsic dimension is $d_i = \dim \mathcal{G}$ since finite groups are zero-dimensional.

| Manifold | Gr (Proj) | Gr (Vec) | St (Matrix) | St (Vec) | Flag (Vec) | Pauli |
|---|---|---|---|---|---|---|
| Int dim $d_i$ | 6-24 | 2-36 | 9-434 | 2-65 | 3-12 | 4-25 |
| Amb dim $d_a$ | 25-900 | 3-924 | 10-660 | 7-960 | 9-300 | 32-1250 |

Table 1: Table detailing the range of the datasets utilized.

**Embedding QuIIEst manifolds**  Our embeddings are constructed using the Gilmore-Perelomov coherent-state method Perelomov (1977; 1986); Zhang et al. (1990), vectorization of matrices, and combinations thereof. The coherent-state method falls into the domains of differential geometry and harmonic analysis, but was discovered independently in mathematical physics. In that context, points in a given homogeneous space parameterize "coherent" quantum states — the states that are closest, in both a technical and a heuristic sense, to the states a quantum system can assume in the classical limit. Such states are typically parameterized by the sphere or complex plane, but coherent states on more general manifolds are relevant to quantum information, quantum metrology Holevo (2011) and, in the case of Grassmannians, the quantum Hall effect Calixto & Pérez-Romero (2014).

The coherent-state method involves choosing a reference vector which is invariant only under the action of the subgroup $\mathcal{H}$. Then, any action of an element of the group $\mathcal{G}$ gives a representation of $\mathcal{G}/\mathcal{H}$. For example, for the two-sphere, the reference vector can be chosen as the north pole. Notice that an $SO(2)$ rotation around the **z**-axis leaves the north pole invariant. Applying the usual Euler-angle representation for general $SO(3)$ rotation yields the 2-sphere. This is explicitly done in Appx. D.1.1. The same manifold can have different embeddings depending on the choice of subgroup and reference vector.

## 4 PROPOSED DATASETS

Table 1 contains a summary of the advantages of our manifolds as compared to other benchmarks. We overview the manifold families below and present explicit constructions in Appx. D. Information about licensing, maintenance and dataset release can be found in Appx. A.

**Stiefel manifolds**  Stiefel manifolds are the closest relatives of spheres out of all QuIIEst manifolds. Real Stiefel manifolds are parameterized by quotients of the form $O(n)/O(n-k)$ for $n \geq k \geq 1$ James (1977), while real spheres can also be understood as the quotients $O(n)/O(n-1)$.

A common alternative definition is the set of $n \times k$ real matrices $X$ satisfying $X^T X = \mathbb{I}$, where $T$ is the transpose map, and where $\mathbb{I}$ is the appropriately sized identity matrix. In this way, one can interpret the manifolds as all possible isometries of $k$-dimensional space into $n$ dimensions. We note that the topology of general Stiefel manifolds is quite different from that of spheres James (1977).

The QuIIEst benchmark includes two different embeddings of Stiefel manifolds. The first, called "St (Matrix)", is a simple vectorization of the matrices $X$. The second, called "St (Vec)", is a mixture of the Gilmore-Perelomov coherent-state method and a vectorization of a matrix.

| IDE Method | Gr (Proj) | Gr (Vec) | Flag (Vec) | St (Matrix) | St (Vec) | Pauli | Average (QuIIEst) |
|---|---|---|---|---|---|---|---|
| lPCA | 1.5492 | 0.0700 | 3.5703 | 0.6210 | 1.4899 | 0.7491 | 1.3416 |
| MLE | 0.1758 | 0.0474 | 0.3065 | 0.4613 | 0.7327 | 0.3913 | 0.3525 |
| CorrInt | 1.0068 | 0.6832 | 4.0588 | 0.7366 | 0.7470 | 0.7570 | 1.3316 |
| TwoNN | 0.0731 | 0.0713 | 0.1970 | 0.4536 | 0.4376 | 0.4456 | **0.2797** |
| ABID | 0.3571 | 0.1858 | 0.4538 | 0.4588 | 0.6182 | 0.0956 | 0.3616 |
| DANCo | 0.3090 | 0.1558 | 5.8647 | 0.7446 | 1.0756 | 1.0692 | 1.5365 |
| **Average** | 0.5785 | **0.2023** | 2.4085 | 0.5793 | 0.8502 | 0.5846 | **0.8672** |

Table 2: Mean relative error $|\delta|$ for various methods on QuIIEst manifolds. A higher value indicates worse performance. We see that the vector embedding of Grassmannian consistently has low error for all methods, while TwoNN typically performs the best on all manifolds. Note however that the native `scikit-dimension` implementation of TwoNN often fails to return an estimate.

**Grassmannians** Grassmannians, or Grassmann manifolds, are defined as $\mathrm{Gr}(k, \mathbb{R}^n) \cong \mathrm{O}(n)/\mathrm{O}(n-k) \times \mathrm{O}(k)$ Lee (2012) and can be thought of as quotients of Stiefel manifolds by an extra $\mathrm{O}(k)$ subgroup. Grassmannians have a long history in ML Zhang et al. (2018); Bendokat et al. (2020). A simple example of one is the real projective plane, $\mathbb{R}P^2 \cong \mathrm{Gr}(1, \mathbb{R}^3)$, which is the sphere $S^2$ with antipodal points identified.

The QuIIEst benchmark includes two different embeddings of Grassmann manifolds. The first is based on the interpretation of the Grassmannian as a space of subspaces. The quotient of the Stiefel manifold by the extra $O(k)$ subgroup identifies two isometries related by a basis change as equivalent. This implies that Grassmannians parameterize all distinct $k$-dimensional subspaces of $n$-dimensional space. We represent a point on the Grassmannian by an $n$-dimensional projector onto a $k$-dimensional subspace, which yields our "Gr (Proj)" embedding when written as an $n^2$-dimensional vector.

The second embedding, called "Gr (Vec)", is based on the coherent-state method and allows for an ambient dimension as low as $\binom{n}{k}$. It is also known as the Plücker embedding Lee (2012); Bendokat et al. (2020).

**Flag manifolds** Flag manifolds generalize Grassmannians to arbitrary *sets* of $n$-dimensional vectors. A $t$-flag manifold is defined as the manifold described by the quotient space $\mathrm{O}(n)/\mathrm{O}(k_1) \times \mathrm{O}(k_2) \times \cdots \times \mathrm{O}(k_{t+1})$, with the constraint $\sum_{i=1}^{t+1} k_i = n$. Our benchmark contains the "Flag (Vec)" embedding of the $t = 2$ case, which is based on the coherent-state method.

**Pauli quotients** This homogeneous space family of real dimension $2n^4$ consists of quotients by a discrete subgroup. It is of the form $\mathrm{U}(n)/\mathcal{P}_n^\star$, where $\mathcal{P}_n^\star$ is a finite subgroup of the unitary group that is defined in Appx. D and that is closely related to the Pauli (a.k.a. Heisenberg-Weyl) group. The corresponding embedding, called "Pauli", is constructed using the coherent-state method with the help of recent results in quantum information theory Bittel et al. (2025).

**Fractals** The definition of manifolds entails that the ID is *the same at every point*. In Appx. H, we discuss ID for fractals, which do not satisfy this definition and which are consequently not included directly in the QuIIEst benchmark. In particular, we discuss Hofstadter's butterfly Hofstadter (1976) as an example of a fractal curve inspired from quantum physics.

Since previous QuIIEst manifolds have the same ID at all points, local ID estimators should return estimates that are not too different at different points — which can be quantitatively measured by the ratio $\sigma(\hat{d}_i)/\langle \hat{d}_i \rangle$ of the standard deviation to the mean ID — and vice-versa for non-manifolds like the Hofstadter's butterfly discussed above. We only include IDE methods which can return fractional estimates of ID. We observe that ABID is a particularly good choice for such non-manifolds, details of which are outlined in the appendix mentioned above.

These manifolds may appear exotic, but they are not as far-removed from real-world data as it seems. Grassmannians Yataka et al. (2023); Huang et al. (2018) and Stiefel Wang et al. (2022); Massart & Abrol (2023) manifolds have been studied in ML before, with the former relevant to airfoil design

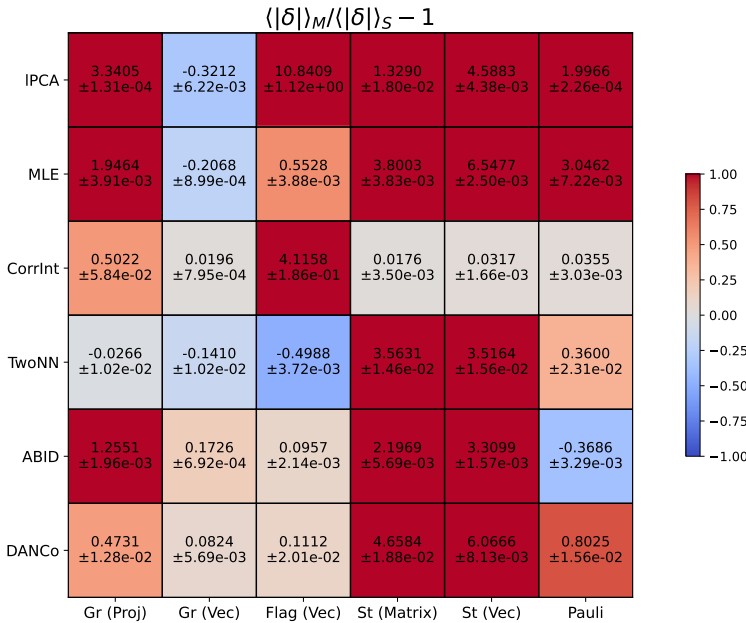

Figure 2: Comparison of the quantity $\langle|\delta|\rangle_M/\langle|\delta|\rangle_S - 1$, where the numerator is the average of the absolute value of the relative error $|\delta|$ over all instantiations of a given manifold family $M$, while the denominator is the corresponding average over all sphere embeddings with the same intrinsic and ambient dimensions. This relative comparison shows that tested methods tend to perform much *worse* against our manifolds than against spheres with the same dimensions. Interestingly, we observe a positive score with a change in embedding of the Grassmannian, hinting that method accuracy depends on the type of embedding. Due to high computational time, we choose a smaller range of hyper-parameter sweeps for DANCo. A 1-$\sigma$ sampling error is reported here after the $\pm$ sign.

Doronina et al. (2022). They are examples of manifolds derived from Lie group symmetries, which have approximate discretized counterparts for real data Goodfellow et al. (2016). In these works, the manifolds typically serve as the base spaces on which learning problems are set up, utilizing flow maps, encoder representations, and coordinate charts to solve the problem more efficiently. To the best of our knowledge, we leverage these manifolds for IDE for the first time.

## 5 RESULTS

### 5.1 METHODS TESTED

We choose a few standard representative IDE methods of different flavors for testing on our benchmark. These are linear methods like linear subspace projection (lPCA Cangelosi & Goriely (2007); Fukunaga & Olsen (1971); Fan et al. (2010)) and non-linear methods such as maximum likelihood estimation (MLE Levina & Bickel (2004); Haro et al. (2008)), fractal dimension estimation (CorrInt Grassberger & Procaccia (1983); Camastra & Vinciarelli (2002)), distribution of measure (TwoNN Facco et al. (2017)), concentration of measure (DANCo Ceruti et al. (2012)), and angle-based moments (ABID Thordsen & Schubert (2022)). We review these methods, including their implementation, in Appx. E.

Given a manifold embedded into a space with ambient dimension $d_a$, we define the *relative error* $\delta$,

$$\delta := \frac{\hat{d}_i}{d_i} - 1 \in \left[-1, \frac{d_a}{d_i} - 1\right] \qquad \text{(relative error)} . \qquad (1)$$

Here, $\hat{d}_i$ is an estimated quantity, while $d_i$ is the ground-truth ID. Note that $\delta < 0$ implies that the method underestimates the ID, while $\delta > 0$ indicates an overestimation for the ID up to the ambient dimension. Average performance of methods on QuIIEst manifolds is summarised in Table 2, in which we list $|\delta|$ so as to compare over- and under-estimation on the same footing.

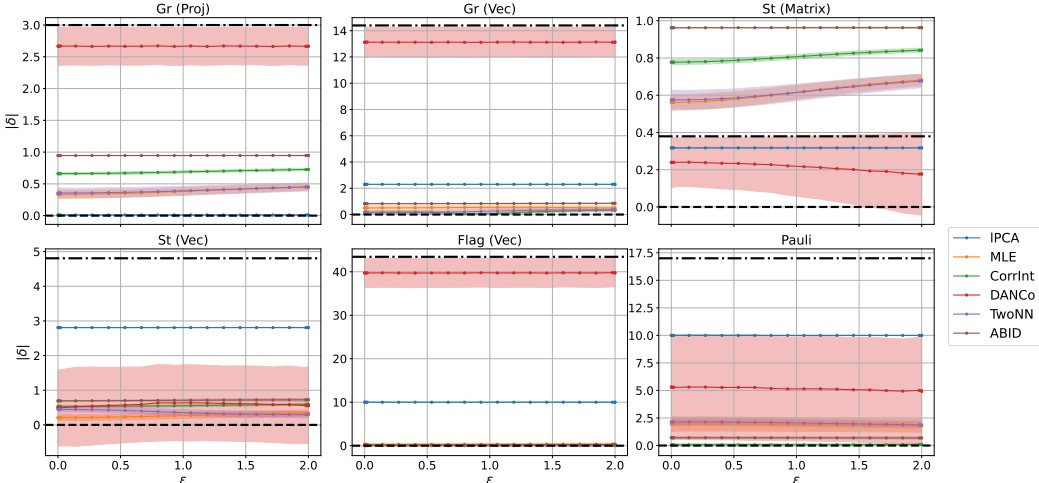

Figure 3: Effect of distortion: We plot the relative error $\langle|\delta|\rangle$ as a function of the parameter $\epsilon$, which is a direct measure of anisotropy. Except for St (Matrix), most methods show negligible change as $\epsilon$ is increased.

## 5.2 COMPARISON WITH OTHER BENCHMARKS

We undertake a comprehensive evaluation of IDE methods on QuIIEst. We perform hyper-parameter sweeps and, for each hyper-parameter combination, compare the performance of different IDE methods on QuIIEst to other IDE benchmarks. We notice that the chosen methods are almost always worse at estimating the ID for our manifold embeddings, with the notable exception being the embedding "Gr (Vec)". For brevity, we present the relative result on spheres here in the main text, cf. Fig 2. The reader is referred to Appx. F for a comparison to other manifolds. We emphasize that the comparison is made across a range of scales, accessible by our computational resources, for manifolds with small ID. The details of the manifolds chosen are discussed in F.

## 5.3 ANISOTROPIC DISTORTIONS : "SQUEEZING"

We now test IDE methods on distorted versions of QuIIEst manifolds, naturally obtained by amending the coherent-state method with generalized "spin squeezing" effects from quantum optics. Distorted manifolds have non-uniform curvature and density, but the same ID. They are obtained by applying a fixed random diagonal matrix to the manifold vectors. The strength of distortion is governed by a parameter $\epsilon$, and we generate each diagonal entry by sampling uniformly from $[1 - \frac{\epsilon}{2}, 1 + \frac{\epsilon}{2}]$. The performance of methods is mostly unchanged upon distortion of the underlying manifolds, cf. Fig 17a. Some methods show a slight degradation in performance, but the change is minimal and within the error margin. By contrast, the methods performed significantly worse on distorted spheres than on our distorted manifolds, cf. App G.2.

## 5.4 DISTORTION THROUGH ADDITIVE NOISE

We perform experiments by perturbing our data from particular manifolds with additive noise, i.e. $\mathbf{x} \to \mathbf{x} + \epsilon$, where $\epsilon$ is sampled from a gaussian distribution $\mathcal{N}(\mathbf{0}, \boldsymbol{\Sigma})$ where $\boldsymbol{\Sigma}$ either the identity matrix $\mathbf{I}_{d_a}$ (*isotropic*), a diagonal matrix $\Lambda$ with elements chosen from $\mathcal{U}(0, \frac{2}{d_a})$ (*uncorrelated*), or a random positive definite matrix $uu^T$ (*anisotropic*), where the elements of $u \in \mathbb{R}^{d_a \times d_a}$ are chosen from $\mathcal{N}(0, 1)$. We then explicitly set $\mathrm{Tr}\,\Sigma = 1$ by the transformation $\Sigma \to \Sigma' = \Sigma/(\mathrm{Tr}\,\Sigma)$. The results are summarized in Fig. 4.

Most methods deviate in their estimations when $\sigma^2 \sim ||\mathbf{x}||_2^2 = \mathcal{O}(1)$. Rather curiously, we observe flat lines, indicating that, for certain methods, the data cloud is indistinguishable from pure noise. We also observe an improvement in IDE performance for certain manifold-method combinations, suggesting a certain regularizing effect emerging from the noise.

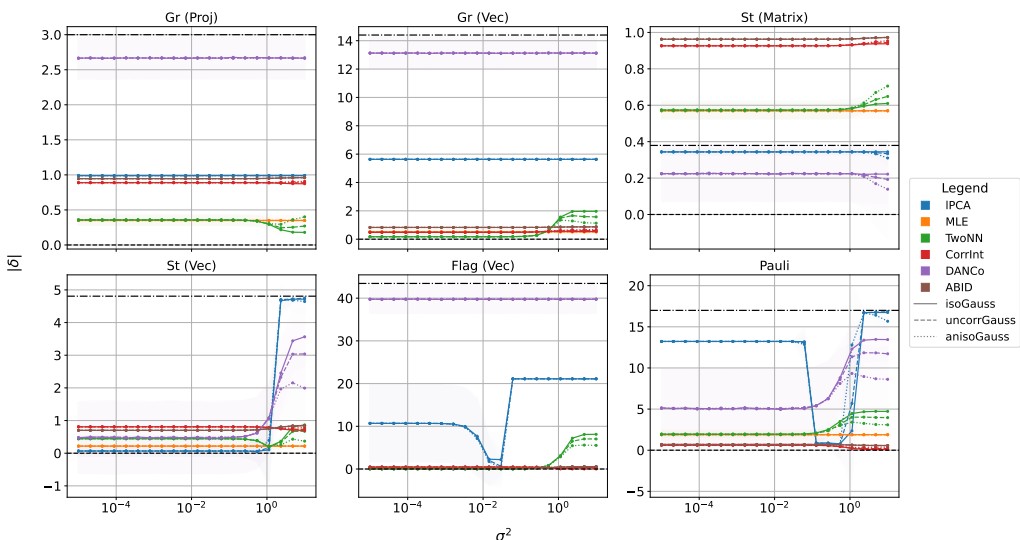

Figure 4: Effect of additive noise. We report IDE performance when the uncorrupted data $\mathbf{x} \rightarrow \mathbf{x} + \epsilon$ where $\epsilon \sim \mathcal{N}(\mathbf{0}, \sigma^2 \Sigma)$ with $\Sigma$ chosen to be proportional to the identity (isotropic), diagonal (uncorrelated) or a complete random symmetric matrix (anisotropic). We then plot the relative error $\delta$ as a function of the noise scale $\sigma^2$. Notice that there is no discernible change in behavior between the different noise types, except in the high noise limit, where the anisotropic noises are consistently underestimated.[a] A 1-$\sigma$ sampling error is plotted.

[a]The figures shown here plot the absolute value of $\delta$, but we numerically confirmed that $\delta$ is smaller.

We observe that in the low to intermediate regime, there is no discernbile change in the behavior of isotropic or anisotropic noise. However, in the high noise limit, we observe that anisotropic noise is always underestimated.

## 5.5 SCALING EXPERIMENTS

**Scaling with data dimensionality** One of the key advantages of QuIIEst manifolds is the independent tuning of the ID and ambient dimensions for the same family of manifolds. This allows us to probe the effect of data dimensionality while sampling from the *same* distribution. We observe that, for fixed sample size and hyper-parameters, the methods progressively become better at estimating the ID as we increase the true ID for most manifolds, with a transition from overestimation at low ID to underestimation at high ID. The notable exception is the vector embedding for the Grassmannian family, with all methods (except ABID) over-estimating the ID after a sufficiently large value. All plots are presented in Appx. G.3.1

**Scaling with sample size** Several arguments Hein & Audibert (2005); Levina & Bickel (2004); Verma (2009); Pestov (2008); Erba et al. (2019) exist to show that reliable ID estimates can be made with a number of samples exponential in the true $d_i$ of the manifold. Hence, the error $\delta$ should decrease with increase in sample size. We observe that this is generally true, but there is no universal convergence value. We detail the results of our investigations in Appx. G.3.2.

## 5.6 RELATION TO REAL DATA

Assuming the manifold hypothesis holds, we conjecture that real-world data has direction-dependent (i.e., anisotropic) curvature — a feature shared by our manifolds. Further distorting the curvature of either our manifolds or MNIST does not lead to a drop in IDE performance, but distorting the initially isotropic curvature of a sphere does. This suggests that our manifolds may be accurately simulating real-world data, at least with respect to this particular feature. Corroborating the need to

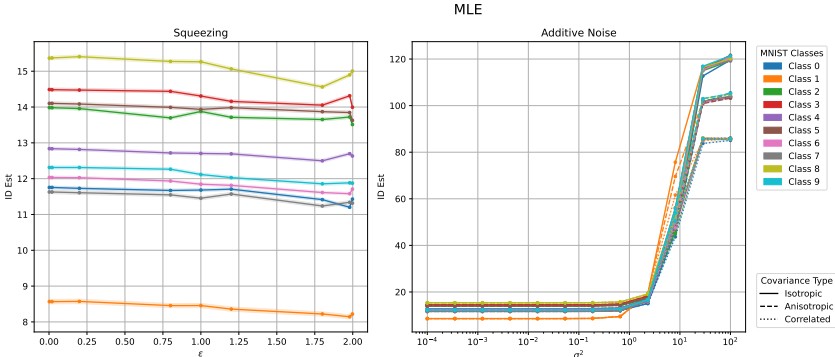

Figure 5: IDE performance for MLE on MNIST when subjected to distortion and noise perturbation. The other plots are presented in Appx. G.6.

use more complex manifolds than spheres, topological properties of MNIST have been studied and are markedly different from that of "simple" manifolds like the sphere Garin & Tauzin (2019).

## 6 ANALYSIS

We analyze IDE performance in terms of different statistical and geometric features of the data.

**Statistical properties** We look at three important features of the data covariance matrix $\mathbf{\Sigma}$: (1) the total variance given by Tr $\mathbf{\Sigma}$, (2) the variance dispersion index (VDI) Watanabe (2022) which is a direct measure of the anisotropy of the data and is given by $\frac{\text{Var}(\lambda)}{(\text{Mean}(\lambda))^2}$ ( $\lambda$ refers to the eigenvalues of $\mathbf{\Sigma}$), and (3) the $\langle R \rangle^2$ value defined as $\frac{1}{d_a^2} \sum_{i,j} \frac{\mathbf{\Sigma}_{ij}^2}{\mathbf{\Sigma}_{ii} \mathbf{\Sigma}_{jj}}$, which captures the inter-component correlation. Note that anisotropy does not imply that components are correlated, but correlated components necessarily imply anisotropy.

We observe that there is a slight negative correlation of performance with anisotropy, the effect being most prominent for the angle-based methods DANCo and ABID, cf. Fig 16c. On the other hand, most methods show almost no correlation with the total variance, except for the angle-based methods. We also observe a slight positive correlation between performance and $\langle R^2 \rangle$. The plots for these results are presented in Appx. G.4.

**Geometric properties** We measure the local mean curvature $H$, local density $\rho$ and a dimensionless parameter $\kappa \equiv \rho/H^{d_i}$. However, we fail to observe any significant dependence on these scalar properties. We outline the details of this investigation in Appx. G.5. We thus conjecture that IDE fails due to the anisotropic curvature of QuIIEst manifolds.

**Role of anisotropic curvature** To leading order, the Euclidean and geodesic distances between two nearby points differ because of the curvature Lang (1995). For manifolds with anisotropic curvature, this difference depends on the points. Nearest-neighbor (NN) IDE methods use Euclidean distances to choose the $k$NNs of a point on the manifold, which can cause systematic bias in the choice of neighborhoods (assuming finite neighbors $k$ and sample size $N$). As such, kNN-based IDE methods should fail for manifolds with anisotropic curvature. This is confirmed by our experiments with the (anisotropic) manifolds of QuIIEst.

**Failure Mode Analysis for lPCA** We conjecture that lPCA fails due to deviation from the expected distribution for manifolds. Recall that lPCA looks at the directions of maximal variance in a local neighborhood. For a manifold, the tangent directions show zero variance. Thus we expect a sharp drop in the relative eigenvalues at the index corresponding to the ID. In Fig. 6, we observe that spheres show this behavior, but distorted spheres and QuIIEst manifolds do not. In particular, this gap sharply decreases as we distort the sphere, while the Grassmanian exhibits a drop at a different

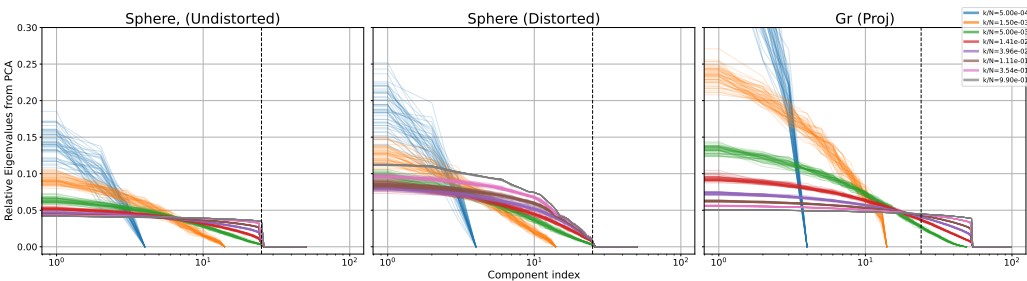

Figure 6: Analysis of relative eigenvalues for lPCA. Recall that the relative eigenvalues represent the fraction of total variance explained by top PCA components. We observe that the relative eigenvalues show a sharp decline at the correct ID value for undistorted spheres, the kink disappearing with distortion. However, for Grassmannian projectors, we observe that the sharp drop does happen but at a different ID. Dashed black lines represent the ID of the manifold.

ID. This trend, however, does not occur for all QuIIEst manifolds, which we describe in detail in Appx. G.10.

We believe that other IDE methods also fail due to systematic deviations introduced by anisotropic curvature, but leave the verification of our prediction to future work.

## 7 CONCLUSION, LIMITATIONS AND FUTURE WORKS

We present QuIIEst— a set of manifold embeddings with complex topological and geometrical structure to be used as a benchmark for intrinsic dimension estimation (IDE). We believe this is an important step in using these IDE methods for the estimation of real-world datasets of unknown ID. Due to constraints on (compute) time and (human) effort, we restrict ourselves to only six IDE methods. Because of this, our analysis on correlation between data properties and IDE performance is limited: we observe weak correlations, but more samples are needed to relate our results to asymptotic estimates of method accuracy. We also provide evidence that QuIIEst resembles real-world data more closely than previous benchmarks due to the anisotropic curvature of the manifolds.

We also notice that IDE methods perform differently on estimating the ID of a manifold embedded using different techniques. Investigating this further may yield more favorable embeddings of real-world data.

We generate embeddings of homogeneous spaces $\mathcal{G}/\mathcal{H}$ for group pairs $\mathcal{H} \subset \mathcal{G}$ using the Gilmore-Peremolov coherent-state method, and outline a simple extension to double coset spaces $\mathcal{K}\backslash\mathcal{G}/\mathcal{H}$ for subgroup $\mathcal{K}$ Albert et al. (2020). Since these spaces need not be manifolds, this extension is a promising route to emulating real-world data not living on a manifold.

Having identified the importance of non-isotropic curvature, we believe that methods should incorporate the (Riemann or Ricci) curvature tensor in their construction. First estimating the ID, then extracting the curvature, and then modifying the ID estimate by using the curvature to determine nearest-neighbor distances may yield a more robust estimation strategy.

Going further, we believe it is possible to extend the coherent state method *directly* to data vectors. For example, given a relevant group of transformations, we can generate new data by applying group elements to a data vector. We hope this will yield concrete connections between the topological and geometrical features of our manifolds and real-world datasets. Since our manifolds have the same ID at every point, one can gauge how much these methods deviate in their IDE at different points. This can serve as a useful diagnostic for the manifold hypothesis.

### DECLARATION OF LLM USAGE

We declare that LLMs were used for minor polishing and formatting of the text, equation and figures, and for correcting grammar.

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

## A    LICENSING, RELEASE, AND DATASET MAINTENANCE

The QuIIEst dataset will be released under the **Creative Commons Attribution 4.0 International (CC BY 4.0)** license, allowing use, modification and redistribution with proper attribution. The dataset consists of embeddings for several quantum-inspired manifolds and contains no reference to any human data or sensitive information.

**Release and Maintenance plan:** The dataset will be released through a public Github repository. As presented in the Supplementary Material, scripts for generating embeddings of the different QuIIEst manifolds will be uploaded, with clear instructions on the sampling process, including but not limited to relevant libraries, necessary computational budget, etc.

Due to memory constraints, we will not be releasing the actual data samples, especially very high-dimensional ones. Instead, we will release the scripts used to generate samples from a particular QuIIEst manifold embedding.

The Github repo will be additionally tracked to allow users to report and fix bugs, additional manifold updates and so on. We will further invite the community to contribute to existing datasets through bug reports, suggestions for improvement and new dataset and feature suggestions. Each update will be accompanied by well-documented release notes, detailing changes and new requirements.

We anticipate integrating QuIIEst with existing IDE libraries, providing additional avenues for long-term maintenance and community contributions. As such, since these are synthetic datasets, the original datasets can be maintained indefinitely. However. through the above practices, we hope to ensure that the dataset remains high-quality, well-maintained and easily accesible to the research community.

**Intended Use:** This dataset is intended primarily for research and benchmarking purposes. In particular, we envision these datasets to be useful and relevant for IDE performance evaluation and aspects of manifold hypothesis. Users are encouraged to cite the dataset and the accompanying paper when reporting results. While the dataset is released under an open license, any use must respect the intended research purpose and proper attribution requirements.

**Ethics and Privacy Standards:** No human subjects or sensitive information are involved in this dataset. The data is entirely synthetic, ensuring full compliance with privacy and ethical standards. Users are encouraged to adhere to best practices in computational research and reproducibility when using the dataset.

## B    EXPERIMENTAL DETAILS

We generated the data with the respective embeddings through custom scripts.

We tested 6 IDE methods — lPCA, MLE, CorrInt, TwoNN, DANCo and ABID. Among these, lPCA was implemented manually, while ABID was implemented via the methodology shared by the authors of Thordsen & Schubert (2022). All other methods were obtained directly from the `scikit-dimension` package. All methods involved computing $k$ nearest neighbors (kNNs) which were pre-computed using the `Nearest Neighbors` module from sklearn.

For comparisons involving QuIIEst and other benchmarks, we ran sweeps for the hyperparameter $k$ and the sample size $N$ for each of the methods. In particular, we performed three types of sweeps - sweeping $N$ logarithmically from 100 to 10000, holding $k$ fixed at 50, sweeping $k$ logarithmically from 10 to 1000 while holding $N$ fixed at 5000 and sweeping $N$ from 100 to 10000 while holding $k/N$ fixed at the values [0.08, 0.1, 0.15, 0.2, 0.5, 0.99]. However, as mentioned in the main text, in order to optimize computational resources, we had to make smaller sweeps for certain IDE-manifold combination. Any other hyper-parameters were held fixed at their default values, after small-scale experiments showed that the effect of these hyper-parameters were not as significant as compared to $k$ and $N$. All runs were performed with 3 different random seeds.

For the scaling experiments, we held $k$ fixed for all runs. The default values were $k = 100(200)$ for lPCA; $k = 100$ for MLE, ABID, TwoNN; $k = 10$ for DANCo; $k_1 = 10, k_2 = 20$ for CorrInt. In the case of $k >= N - 1$, in which case we chose $k = N - 2$; for CorrInt, this was modified as $k_2 = N - 2$ and $k_1 = k_2/2$. We used $k = 100$ for the experiment where we scaled the sample size

918
919
920
$N$, while we used $k = 200$ for the case of scaling with noise. All other hyperparameters were kept at their default values.

921
922
923
Experiments (including data generation) were run on dual AMD 7763 32-core CPUs and took around 600 CPU hours with an approximate total of 10 hours for data generation. Plots and inferences were then made locally with negligible overhead cost.

924
925
926
There are two dominant sources of errors for our plots — spread in the local ID estimates for the entire sample, and the error due to using three different random seeds. The error reported is the mean error obtained for the local estimates, averaged over 3 random seeds.

927
928
929
930
931
932
933
934
935
936
937
938
939
940
941
942
943
944
945
946
947
948
949

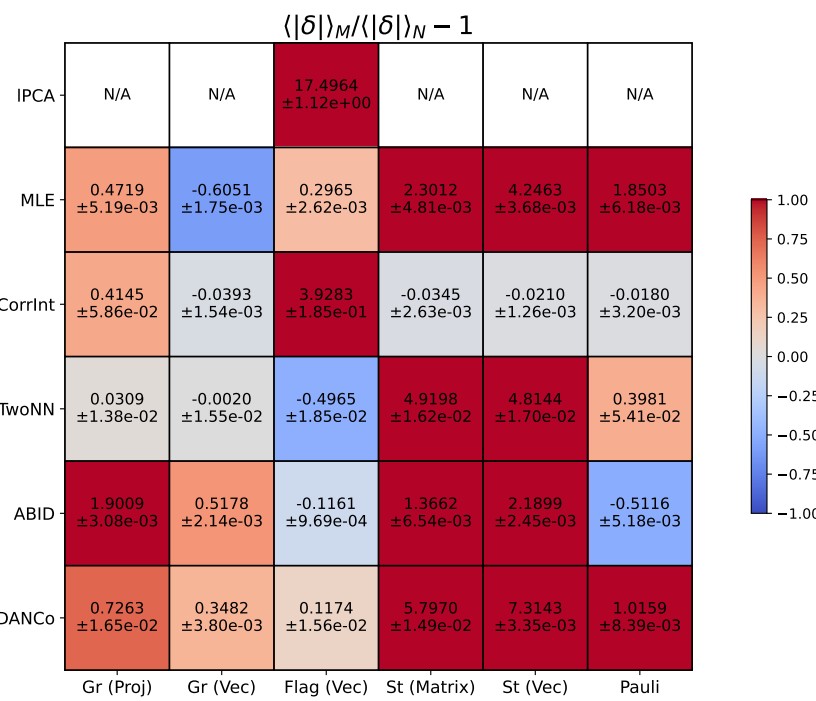

950
951
952
Figure 7: Relative performance of IDE methods on QuIIEst manifolds and the manifold of normal isotropic Gaussian vectors . N/A indicates that the method performs very well on $\mathcal{N}$.

953
954
955

## C  DETAILED BACKGROUND AND RELEVANT WORKS

956
957
958
959
960
961
962
963
964
**The manifold paradigm in ML**  The manifold hypothesis has been a well-known paradigm in the machine learning community Gorban & Tyukin (2018a); Olah (2014); Cayton (2005). This hypothesis has been subject to both theoretical and experimental investigations Fefferman et al. (2013); Goldt et al. (2020); Narayanan & Mitter (2010); Kiani et al. (2024); Brown et al. (2023); Donoho & Grimes (2005); Lee et al. (2003). At the same time, there has been interest in understanding the topological structures of real-world datasets through the field of Topological Data Analysis Carlsson et al. (2008); Zia et al. (2024). There is a separate notion involving manifolds in ML, which seeks to understand the learning process as modification of latent representations of the data manifold, as explored in Magai & Ayzenberg (2022); Ansuini et al. (2019).

965
966
967
968
969
970
971
**Intrinsic Dimension**  The intrinsic dimension (ID) of a manifold is a very useful quantity for several reasons. It is a characteristic property of the manifold, and hence many properties or features of learning problems are dependent on the ID — dimensionality reduction Zhang & Zha (2005); Gashler & Martinez (2011; 2012), exponential scaling of samples with ID Narayanan & Niyogi (2009); Narayanan & Mitter (2010), correlation of generalization capability with ID of data and internal representations Pope et al. (2021); Ansuini et al. (2019); Birdal et al. (2021); Magai & Ayzenberg (2022); Brown et al. (2022) as well as ID being a natural measure of local complexity of the data Kamkari et al. (2024).

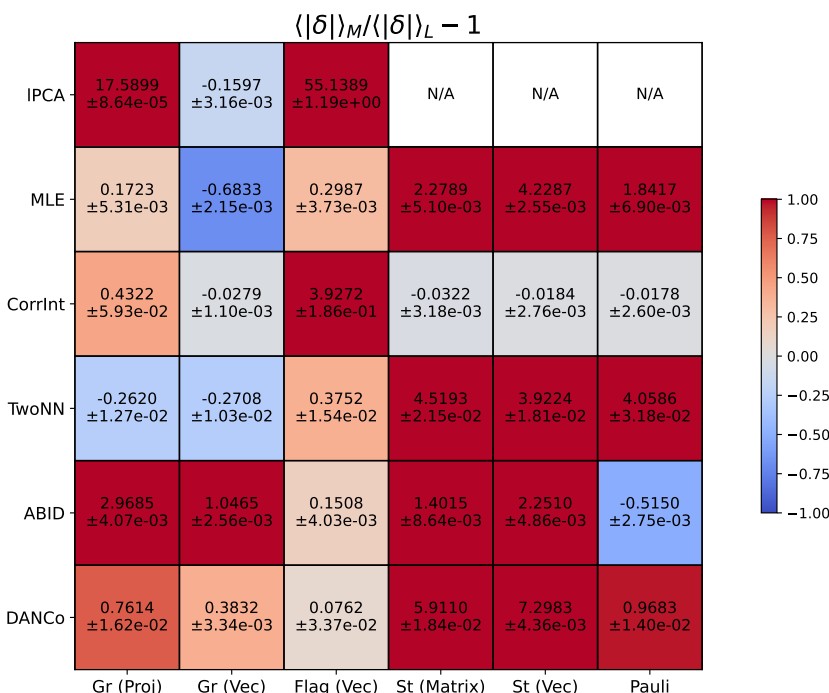

Figure 8: Relative performance of IDE methods on QuIIEst manifolds and the manifold of affine linear nullspace $\mathcal{L}$. N/A indicates that the method performs very well on $\mathcal{L}$.

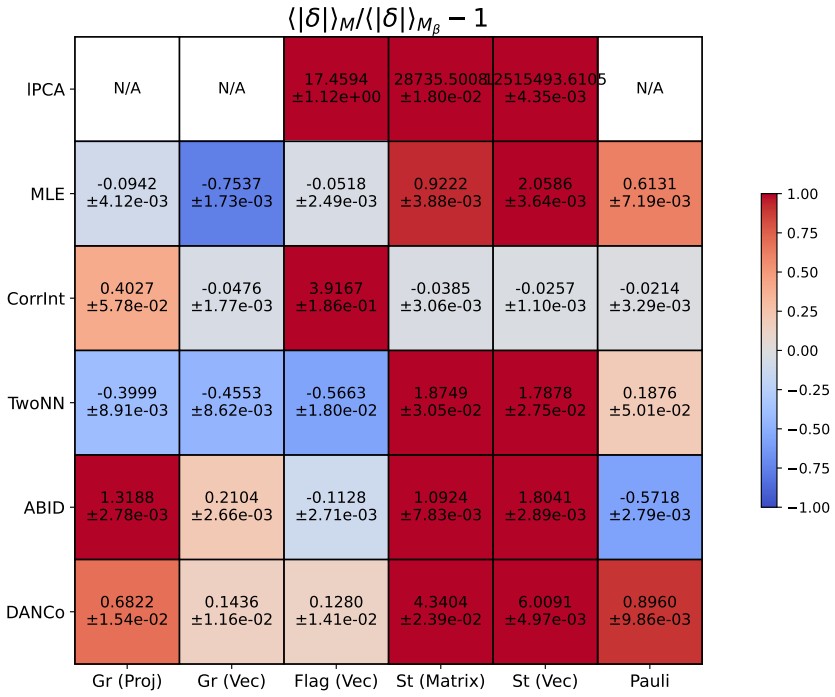

Figure 9: Relative performance of IDE methods on QuIIEst manifolds and the manifold $\mathcal{M}_\beta$. N/A indicates that the method performs very well on $\mathcal{M}_\beta$.

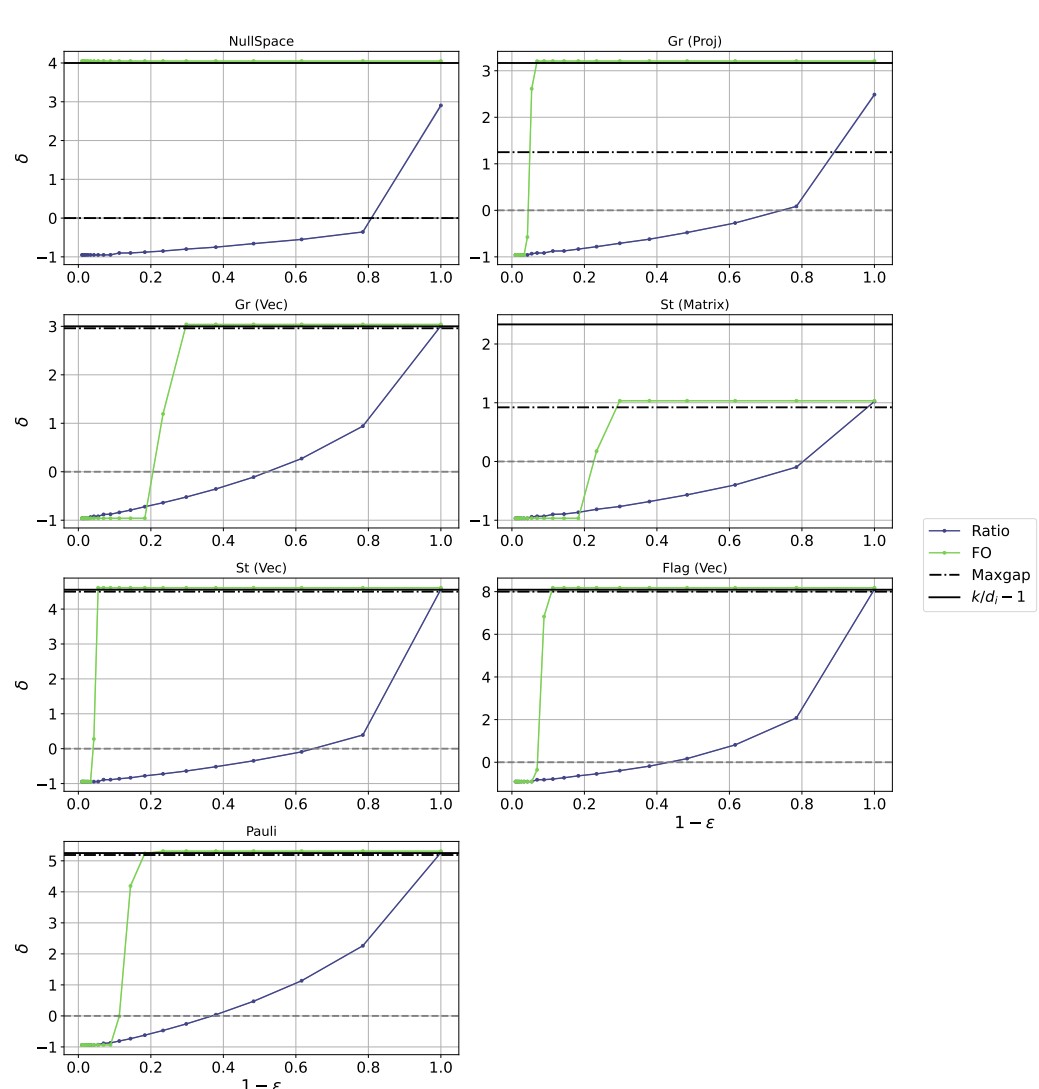

Figure 10: lPCA maxgap compared to lPCA ratio and FO versions. 'FO' usually shows a quicker convergence to lPCA maxgap value as compared to the 'ratio' version.

The importance of ID has thus led to a spate of research on coming up with estimators of ID. These method have different approaches to estimating LID — through various distance-based measures, such as those based on Euclidean distance measure Camastra & Vinciarelli (2002); Levina & Bickel (2004); Gomtsyan et al. (2019); Facco et al. (2017), non-Euclidean geodetic distances Granata & Carnevale (2016) and Wasserstein distances Block et al. (2022); deviations of simplexes Johnsson et al. (2014); angle-based measures Ceruti et al. (2014); Thordsen & Schubert (2022); measures based on generative models Stanczuk et al. (2023); Tempczyk et al. (2021); Kamkari et al. (2024); Zheng et al. (2022) and quantum encoding-based algorithms Candelori et al. (2025). See Campadelli et al. (2015) for a more extensive survey of various ID estimators.

**Benchmarks for IDE**  Camastra (2003) undertook one of the first surveys for ID estimators, outlining different datasets used for benchmarking ID estimators known at the time. Hein & Audibert (2005) constructed a series of manifolds to benchmark their ID estimator, including hyperspheres, isotropic Gaussian vectors, the Möbius strip, etc. However, the authors acknowledged the problem with evaluating ID estimators for MNIST, and attempted to construct datasets with variable ID by applying transformations such as translation, rotation, etc. on MNIST images. Rozza et al. (2012) drew on these ideas and constructed several complicated manifolds, devising means of embedding data into higher dimensions without linear isometries. Campadelli et al. (2015) took this further and proposed a benchmark with several such manifolds. A very interesting technique to benchmark ID estimators on distributions different from these manifolds was proposed in Pope et al. (2021). In their paper, the authors propose using GANs with certain restriction on the dimension $\bar{d}$ of the latent noise vectors to generate images, whose ID was then bounded from above by $\bar{d}$. However, this method suffers from an obvious problem in that there is no ground-truth ID for the generated data.

In addition, most ID estimators have a plethora of synthetic datasets on which they perform their zeroth-order evaluations - but most of them are low-dimensional, "simple" manifolds, whereas real-world data is more complicated and comes from vastly different distributions. There are some attempts to artificially induce complexity in these manifolds, examples of which are highlighted above. Scikit has compiled some of these manifolds into their python library Bac et al. (2021). However, it contains only a few topologically non-trivial manifolds such as the Möbius strip, Swiss roll, helixes, etc. which do not admit natural embeddings for arbitrary intrinsic dimensions. Previous attempts to construct physics-inspired datasets for IDE evaluation involved nonlinear dynamical systems. These include the Santa Fe datasets Weigend & Gershenfeld (1993), DSVC1 dataset Camastra & Filippone (2009) and strange attractors Grassberger & Procaccia (1983). However, the true ID in these cases is not always known a priori, which makes it difficult to use as a benchmark. It is with this background that we propose QuIIEst- an infinite family of manifolds which offer multiple natural notions of embedding in higher-dimensional space. In addition, we propose a recipe to construct such manifolds through the concept of homogenous spaces and coherent states.

**Quantum and ML**  Any quantum algorithm can be simulated on a classical computer, but in a non-scalable way since the dimension of the underlying quantum state space increases exponentially in the number of quantum bits. Quantum-inspired algorithms Cerezo et al. (2022); Arrazola et al. (2020); Gharehchopogh (2022) leverage the advantages of quantum algorithms on existing classical hardware in a scalable way. A well-known example is the quantum-inspired recommendation algorithm of Ewin Tang Tang (2019), which can be thought of as a "de-quantized" version of an earlier quantum recommendation algorithm Kerenidis & Prakash (2016). Various other instances have recently been developed in this nascent subfield, e.g., in Refs. Huynh et al. (2023); Ding et al. (2022).

We also leverage tools from quantum mechanics, but to develop a benchmark instead of an algorithm. Specifically, we use Gilmore-Perelomov coherent states Perelomov (1977; 1986); Zhang et al. (1990) to construct parameterized homogeneous spaces. Coherent states are the closest analogue of a classical state in a quantum system and have been used in high-energy physics, quantum optics, and, most recently, in quantum error correction. We further distort coherent-state spaces in a way akin to generalized spin squeezing, which is useful for measuring signals along certain directions in quantum metrology.

# D   DETAILED THEORY FOR QUIIEST MANIFOLDS

We present a brief overview of the manifolds included in QuIIEst. The interested reader is referred to the references herein for a more in-depth discussion.

All manifolds included in the QuIIEst benchmark are parameterized by quotients $\mathcal{G}/\mathcal{H}$, where $\mathcal{G}$ is a Lie group (i.e., a group that can also be considered as a manifold), and where $\mathcal{H}$ is one of its subgroups. Such spaces are called homogeneous spaces, and they are examples of quotient spaces. When $\mathcal{H}$ is continuous subgroup, the intrinsic dimension of such quotient spaces is $d_i = \dim \mathcal{G} - \dim \mathcal{H}$. When $\mathcal{H}$ is finite, $\dim \mathcal{H} = 0$, and therefore the intrinsic dimension is equal to the dimension of $\mathcal{G}$.

Our embeddings are constructed using the Gilmore-Perelomov coherent-state method Perelomov (1977; 1986); Zhang et al. (1990), vectorization of matrices, and combinations thereof. We review Gilmore-Perelomov coherent-states in Section D.1 and use them to build the QuIIEst manifolds in Section D.2.

## D.1   GILMORE-PERELOMOV EMBEDDINGS

Note: This text has been expanded and subsection D.1.1 has been added during rebuttal.

Given the groups $\mathcal{H} < \mathcal{G}$ and some desired ambient dimension $d_A$ in which we wish to embed the manifold $\mathcal{M} = \mathcal{G}/\mathcal{H}$, we must first construct an orthogonal representation $\Lambda$ of $\mathcal{G}$[1]. Define $\Lambda|_{\mathcal{H}}$ to be the representation restricted to $\mathcal{H}$. For compact Lie groups, all irreducible representations (irreps) appear in the isotypic decomposition of tensor-product representation (https://mathoverflow.net/users/297/david-e speyer). Our starting point will therefore be to consider representations of the form $\Lambda(g) = g^{\otimes t}$, where $g$ is in the defining irrep, but the recipe can be generalized to other representations.

The projector onto the trivial irreps in the isotypic decomposition of a representation $\Lambda$ is $P(\Lambda) = \int_{\mathcal{G}} \Lambda(g)\, dg$, where $dg$ denotes the unique unit-normalized Haar measure on $\mathcal{G}$. We suppose that A). $P(\Lambda|_{\mathcal{H}}) > P(\Lambda)$, meaning that there are nontrivial irreps of $\mathcal{H}$ contained in $\Lambda$, and B). that there are no subgroups $\mathcal{H} < \mathcal{K} < \mathcal{G}$ with $P(\Lambda|_{\mathcal{K}}) > P(\Lambda|_{\mathcal{H}})$.

Given the above assumptions, there exists at least one unit vector $|\mathcal{H}\rangle$ that lies in the image of $P(\Lambda|_{\mathcal{H}})$ and does not lie in the image of $P(\Lambda|_{\mathcal{K}})$ for any $\mathcal{K} > \mathcal{H}$ (including $\mathcal{K} = \mathcal{G}$). By construction, $|\mathcal{H}\rangle$ lies in a trivial irrep of $\Lambda|_{\mathcal{H}}$, and therefore $\Lambda(h) |\mathcal{H}\rangle = |\mathcal{H}\rangle$ for all $h \in \mathcal{H}$. Moreover, due to assumption (B), all other elements of $\mathcal{G}$ do not leave $|\mathcal{H}\rangle$ invariant. Thus, because any $g \in \mathcal{G}$ can be uniquely written as $g = ah$ for a subgroup element $h \in \mathcal{H}$ and coset representative $a \in \mathcal{M} = \mathcal{G}/\mathcal{H}$, the mapping

$$|\psi_{\mathcal{M}}(g)\rangle = \Lambda(g) |\mathcal{H}\rangle = \Lambda(a)\Lambda(h) |\mathcal{H}\rangle = \Lambda(a) |\mathcal{H}\rangle \tag{2}$$

is a well-defined, injective mapping between $\mathcal{M}$ and $\mathbb{R}^{d_A}$ or $\mathbb{C}^{d_A}$ (see footnote 1). In fact, the embedding is injective into the real or complex sphere $\Omega_{d_A}$.

This mapping is equivariant – $\Lambda(g_1) |\psi_{\mathcal{M}}(g_2)\rangle = |\psi_{\mathcal{M}}(g_1 g_2)\rangle$. It is, however, not in general isometric – this depends on the metric that is chosen on $\mathcal{M}$. In particular, if the metric $g$ on $\mathcal{M}$ is induced from the Cartan-Killing metric on $\mathcal{G}$, then $\psi_{\mathcal{M}} \colon (\mathcal{M}, g) \to (\Omega_{d_A}, g_\Omega)$ will not be isometric, where $g_\Omega$ is the metric on the sphere. Of course, we could instead choose $g$ to be the metric induced by the embedding; that is, the pullback $g = \psi_{\mathcal{M}}^* g_\Omega$. In this case, the embedding is isometric by construction.

Finally, $\psi_{\mathcal{M}}$ is an immersion simply because its derivative is everywhere injective. Specifically, suppose that $A$ is a Lie algebra element so that $g(s) = g(0)e^{As}$ is a curve in $\mathcal{M}$. The embedded tangent space elements are then identified with $\frac{d}{ds}\psi_{\mathcal{M}}(g(s))|_{s=0}$.

## D.1.1   EXAMPLE OF THE COHERENT STATE METHOD

We use the Perelomov coherent state method to parametrize points on the sphere:

$$\mathbb{S}^2 \cong SO(3)/SO(2)$$

---

[1]When embedding into $\mathbb{R}^{d_A}$, the representation should be orthogonal; when embedding into $\mathbb{C}^{d_A}$, the representation should be unitary.

**Step 1.** Pick a fiducial state: We choose the north pole which has the coordinates:

$$|\psi_0\rangle = \begin{bmatrix} 0 \\ 0 \\ 1 \end{bmatrix}$$

**Step 2. Representations of the groups:**

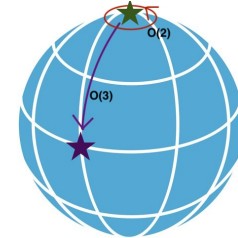

Figure 11: 2-sphere is O(3)/O(2).

- A general rotation $g \in SO(3)$ is parametrized by Euler angles:

$$g(\alpha, \beta, \gamma) = R_z(\alpha) R_y(\beta) R_z(\gamma)$$

  where

$$R_z(\theta) = \begin{bmatrix} \cos\theta & -\sin\theta & 0 \\ \sin\theta & \cos\theta & 0 \\ 0 & 0 & 1 \end{bmatrix}, \quad R_y(\theta) = \begin{bmatrix} \cos\theta & 0 & \sin\theta \\ 0 & 1 & 0 \\ -\sin\theta & 0 & \cos\theta \end{bmatrix}$$

- The stabilizer subgroup $H = SO(2) \subset SO(3)$ consists of rotations about the $z$-axis:

$$h(\gamma) = R_z(\gamma)$$

**Step 3. Invariance of the fiducial state:**

$$R_z(\gamma)|\psi_0\rangle = |\psi_0\rangle \qquad \Rightarrow \qquad |\psi_0\rangle \text{ is invariant under } H$$

Incidentally these are the only rotations which preserve the north pole.

**Step 4. Coherent state construction:**

$$|\psi_g\rangle = g(\alpha, \beta, \gamma)|\psi_0\rangle$$

Apply each part sequentially:

$$R_z(\gamma)|\psi_0\rangle = |\psi_0\rangle, \quad R_y(\beta)|\psi_0\rangle = \begin{bmatrix} \sin\beta \\ 0 \\ \cos\beta \end{bmatrix}, \quad R_z(\alpha)\begin{bmatrix} \sin\beta \\ 0 \\ \cos\beta \end{bmatrix} = \begin{bmatrix} \sin\beta\cos\alpha \\ \sin\beta\sin\alpha \\ \cos\beta \end{bmatrix}$$

**Final result:**

$$\boxed{|\psi_{\alpha,\beta}\rangle = \begin{bmatrix} \sin\beta\cos\alpha \\ \sin\beta\sin\alpha \\ \cos\beta \end{bmatrix}}$$

This parametrizes the full sphere $\mathbb{S}^2$ in the usual polar coordinate form.

### D.1.2 EXTENSION TO NATURAL DATA

The notion of group representations and fiducial vectors can be easily extended to their discrete counterparts. If there exists a natural vectorization of the data, one can make the correspondence $\mathbf{x} \to |\mathcal{H}\rangle$ by setting $\langle e_i|\mathcal{H}\rangle = x_i$ where $|e_i\rangle$ denotes some standard basis spanning a $d$-dimensional Hilbert space. One can then use these to generate an embedding for a homogenous space as outlined above.

As an example, consider the number '1' in MNIST. The image admits translation invariance, and so one can use any standard image representing '1' as the fiducial vector to represent a group $\mathcal{G}/\mathcal{T}$ where $\mathcal{T} < \mathcal{G}$ represents discrete translations.

### D.2 EMBEDDING MANIFOLDS INTO EUCLIDEAN SPACE

Throughout this section, let $\{|1\rangle, \ldots, |n\rangle\}$ be the standard (unit-vector) basis of $\mathbb{R}^n$. Furthermore, for a matrix $X$, define $\vec{X}$ to be the vectorization of $X$ – that is, the vector obtained by stacking the columns.

| Family | Symbol | Quotient space | $d_i$ | QuIIEst embedding | $d_a^{\min}$ |
|---|---|---|---|---|---|
| Stiefel manifold | $\mathrm{St}(k, \mathbb{R}^n)$ | $\frac{\mathrm{O}(n)}{\mathrm{O}(n-k)}$ | $nk - \frac{1}{2}k(k+1)$ | St (Matrix) | $nk$ |
| | | | | St (Vec) | $\binom{n}{k} + k^2$ |
| Grassmannian | $\mathrm{Gr}(k, \mathbb{R}^n)$ | $\frac{\mathrm{O}(n)}{\mathrm{O}(n-k)\mathrm{O}(k)}$ | $k(n-k)$ | Gr (Proj) | $n^2$ |
| | $\mathrm{Gr}^\star(k, \mathbb{R}^n)$ | $\frac{\mathrm{O}(n)}{\mathrm{O}(n-k)\mathrm{SO}(k)}$ | $k(n-k)$ | Gr (Vec) | $\binom{n}{k}$ |
| Flag manifold | $\mathrm{Flag}^\star(k_1, k_2, \mathbb{R}^n)$ | $\frac{\mathrm{O}(n)}{\mathrm{SO}(k_1)\mathrm{SO}(k_2)\mathrm{O}(n-k_1-k_2)}$ | $(k_1+k_2)n$ $-k_1^2-k_2^2-k_1 k_2$ | Flag (Vec) | $\binom{n}{k_1}\binom{n}{k_2}$ |
| Pauli quotient | $\frac{\mathrm{U}(n)}{\mathcal{P}_n^\star}$ | $\frac{\mathrm{U}(n)}{\mathcal{P}_n^\star}$ | $n^2$ | Pauli | $2n^4$ |

Table 3: Table listing manifold families used in the QuIIEst benchmark, their mathematical symbols, their equivalent quotient/homogeneous spaces, and their intrinsic dimensions $d_i$. Here, $\mathrm{U}(n)$ and $\mathrm{O}(n)$ denote the unitary and orthogonal groups in $n$ dimensions, respectively, $\mathrm{SO}(n)$ denotes the special unitary groups, and the group $\mathcal{P}_n^\star$ is defined in Sec. D. The parameter $n \geq 1$ for all rows except the last one, where it is assumed to be prime. The last two columns list the six QuIIEst embeddings and their lowest possible ambient dimensions, $d_a^{\min}$. Any embedding into a given ambient dimension can be further embedded into a space of larger ambient dimension via any isometry. The use of slightly different quotients for our two Grassmannian embeddings yields a lower possible ambient dimension for the latter embedding while maintaining the same intrinsic dimension. Note that the usual flag manifold can be obtained from $\mathrm{Flag}^\star(k_1, k_2, \mathbb{R}^n)$ by letting $\mathrm{SO} \to \mathrm{O}$. The symbol $\binom{a}{b}$ is the binomial coefficient.

STIEFEL MANIFOLDS AND THE "ST (MATRIX)" EMBEDDING

The Stiefel manifold is $\mathrm{St}(k, \mathbb{R}^n) = \left\{ X \in \mathbb{R}^{n \times k} \mid X^T X = \mathbb{I} \right\}$ James (1977); Bendokat et al. (2020). This easily embeds into $\mathbb{R}^{nk}$ via the map $X \mapsto \vec{X}$. This yields the "St (Matrix)" embedding in the QuIIEst dataset.

A more interesting embedding will be used as an intermediate step to derive the Grassmannian embedding used in QuIIEst and is as follows. We will view $\mathrm{St}(k, \mathbb{R}^n)$ as $\mathrm{O}(n)/\mathrm{O}(n-k)$. Then a point $X \in \mathrm{St}(k, \mathbb{R}^n)$ is given by the first $k$ columns of the corresponding orthogonal matrix. Using the notation from Section D.1, we let $\mathcal{G} = \mathrm{O}(n)$ and $\mathcal{H} = \mathrm{O}(n-k)$, and define the representation $\Lambda \colon \mathcal{O} \mapsto \mathcal{O}^{\otimes k}$ for $\mathcal{O} \in \mathrm{O}(n)$. An obvious choice for the state $|\mathcal{H}\rangle$ is $|1\rangle \otimes \cdots \otimes |k\rangle$. Given $\mathcal{O} \in \mathrm{O}(n)$, this yields the embedding

$$|\psi_{\mathrm{St}}(\mathcal{O})\rangle = \Lambda(\mathcal{O}) |\mathcal{H}\rangle \tag{3}$$

$$= \mathcal{O} |1\rangle \otimes \mathcal{O} |2\rangle \otimes \cdots \otimes \mathcal{O} |k\rangle \tag{4}$$

$$= \sum_{j_1, \ldots, j_k=1}^{n} \mathcal{O}_{j_1,1} \ldots \mathcal{O}_{j_k,k} |j_1\rangle \otimes \cdots \otimes |j_k\rangle \tag{5}$$

$$= |\mathcal{O}_{*,1}\rangle \otimes \cdots \otimes |\mathcal{O}_{*,k}\rangle , \tag{6}$$

where $|\mathcal{O}_{*,i}\rangle$ denotes the $i^{\mathrm{th}}$ column of $\mathcal{O}$. From this we can easily see that $|\psi_{\mathrm{St}}(\mathcal{O})\rangle$ depends only on the first $k$ columns of $\mathcal{O}$, but will result in a different vector for different elements of the Stiefel manifold. It is therefore an injection of the Stiefel manifold into $\mathbb{R}^{n^k}$.

GRASSMANNIANS AND THE "GR (VEC)" EMBEDDING

Consider the embedding $|\psi_{\mathrm{St}}(\mathcal{O})\rangle$ of the Stiefel manifold from above. We need to slightly modify this to get an embedding of the Grassmannian,

$$\mathrm{Gr}^\star(k, \mathbb{R}^n) \cong \frac{\mathrm{St}(k, \mathbb{R}^n)}{\mathrm{SO}(k)} \cong \frac{\mathrm{O}(n)}{\mathrm{O}(n-k) \times \mathrm{SO}(k)} , \tag{7}$$

| $\mathcal{M}$ | $=$ | $\mathcal{G}/\mathcal{H}$ | $\pi_1(\mathcal{M})$ | $\pi_2(\mathcal{M})$ |
|---|---|---|---|---|
| $\mathrm{Gr}(k, \mathbb{R}^n)$ | $=$ | $\mathrm{O}(n)/\mathrm{O}(n-k) \times \mathrm{O}(k)$ | $\mathbb{Z}_2$ | $\mathbb{Z}_2$ |
| $\widetilde{\mathrm{Gr}}(k, \mathbb{R}^n)$ | $=$ | $\mathrm{SO}(n)/\mathrm{SO}(n-k) \times \mathrm{SO}(k)$ | $0$ | $\mathbb{Z}_2$ |
| $\mathrm{Gr}^\star(k, \mathbb{R}^n)$ | $=$ | $\mathrm{O}(n)/\mathrm{O}(n-k) \times \mathrm{SO}(k)$ | $\mathbb{Z}_2$ | $\mathbb{Z}_2$ |

Table 4: The first two homotopy groups of the three types of "Grassmannians" described below Eq. equation 7.

due to the extra quotient by $\mathrm{SO}(k)$. In particular, we define

$$|\mathcal{H}\rangle = \frac{1}{\sqrt{k!}} \sum_{\sigma \in S_k} \mathrm{sgn}(\sigma) |\sigma(1)\rangle \otimes \cdots \otimes |\sigma(k)\rangle, \tag{8}$$

where $\mathrm{sgn}(\sigma)$ denotes the symmetric group on $k$ elements.

We note that Eq. equation 7 is not the usual definition of the Grassmannian. Indeed, Grassmannian is typically $\mathrm{Gr}(k, \mathbb{R}^n) = \frac{\mathrm{O}(n)}{\mathrm{O}(n-k) \times \mathrm{O}(k)}$, and the oriented Grassmannian is $\widetilde{\mathrm{Gr}}(k, \mathbb{R}^n) = \frac{\mathrm{SO}(n)}{\mathrm{SO}(n-k) \times \mathrm{SO}(k)}$. These two spaces, along with the space in Eq. equation 7, all have the same ID. In future work, we will add the standard and oriented Grassmannian to QuIIEst and further compare the performance of standard ID estimation methods on these three very similar manifolds. In particular, the geometry and topology of these three manifolds are slightly different despite the manifolds themselves being morally very similar, thus yielding an interesting testing ground. For example, we list the first two homotopy groups of these manifolds in Table. 4, which we caluclate using the long exact sequence of homotopy groups induced from the fiber bundle $\mathcal{H} \to \mathcal{G} \to \mathcal{G}/\mathcal{H}$.

Given $\mathcal{H}$, we achieve the embedding

$$|\psi_{\mathrm{Gr}}(\mathcal{O})\rangle = \Lambda(\mathcal{O}) |\mathcal{H}\rangle \tag{9}$$

$$= \frac{1}{\sqrt{k!}} \sum_{\sigma \in S_k} \mathrm{sgn}(\sigma) \mathcal{O} |\sigma(1)\rangle \otimes \cdots \otimes \mathcal{O} |\sigma(k)\rangle \tag{10}$$

$$= \frac{1}{\sqrt{k!}} \sum_{\sigma \in S_k} \sum_{j_1,\ldots,j_k=1}^{n} \mathrm{sgn}(\sigma) \mathcal{O}_{j_1,\sigma(1)} \ldots \mathcal{O}_{j_k,\sigma(k)} |j_1\rangle \otimes \cdots \otimes |j_k\rangle \tag{11}$$

$$= \frac{1}{\sqrt{k!}} \sum_{j_1,\ldots,j_k=1}^{n} \det(\mathcal{O}_{(j_1,\ldots,j_k)}) |j_1\rangle \otimes \cdots \otimes |j_k\rangle, \tag{12}$$

where $\mathcal{O}_{(j_1,\ldots,j_k)}$ denotes the $k \times k$ matrix obtained from $\mathcal{O}$ by taking the first $k$ columns and taking the rows $j_1,\ldots,j_k$, and similarly $\mathcal{O}_{\{j_1,\ldots,j_k\}} = \mathcal{O}_{\mathrm{sorted}(j_1,\ldots,j_k)}$.

Let $\sigma \in S_k$ be the permutation that sorts $(j_1,\ldots,j_k)$. Then $\det \mathcal{O}_{(j_1,\ldots,j_k)} = \mathrm{sgn}(\sigma) \det \mathcal{O}_{\{j_1,\ldots,j_k\}}$. Therefore, we have that

$$|\psi_{\mathrm{Gr}}(\mathcal{O})\rangle = \sum_{\substack{Q \subset \{1,\ldots,n\} \\ |Q|=k}} \det \mathcal{O}_Q \frac{1}{\sqrt{k!}} \sum_{\sigma \in S_k} \mathrm{sgn}(\sigma) |\sigma(Q_1)\rangle \otimes \cdots \otimes |\sigma(Q_k)\rangle \tag{13}$$

$$= \sum_{\substack{Q \subset \{1,\ldots,n\} \\ |Q|=k}} \det \mathcal{O}_Q |Q\rangle, \tag{14}$$

where we defined $Q_i$ to be the $i^{\mathrm{th}}$ element of the set $Q$ (of course this does not make sense generally, but because we are summing over all permutations, it is fine to pick some arbitrary ordering of the set), and we defined the state

$$|Q\rangle = \frac{1}{\sqrt{k!}} \sum_{\sigma \in S_k} \mathrm{sgn}(\sigma) |\sigma(Q_1)\rangle \otimes \cdots \otimes |\sigma(Q_k)\rangle. \tag{15}$$

Notice that we are embedding into $\mathbb{R}^{n^k}$, but a full basis for the space is given by $|Q\rangle$ for all $Q \subset \{1,\ldots,n\}$ with $|Q|=k$. Thus, this embedding in general gives us an embedding into $\mathbb{R}^{d_A}$ for any $d_A \geq \binom{n}{k}$. We denote this by the shorthand "Gr (Vec)" in Table 3.

We note that this embedding is almost the Plücker embedding Bendokat et al. (2020). The Plücker embedding is an embedding of the standard Grassmannian into real projective space, which is the real sphere with antipodal points identified. Above, we are embedding Eq. equation 7 into the real sphere. The reason that $\psi_{\mathrm{Gr}}$ is not an embedding of $\mathrm{O}(n)/\mathrm{O}(n-k) \times \mathrm{O}(k)$ is because the vector $|\mathcal{H}\rangle$ is not invariant under the action $\mathrm{O}(k)$. Instead, it is invariant under the action of $\mathrm{SO}(k)$, and yields a $\pm 1$ phase factor under the action of $\mathrm{O}(k)$. This yields a well-defined embedding of $\mathrm{O}(n)/\mathrm{O}(n-k) \times \mathrm{O}(k)$ into projective space, but further embedding projective space into the sphere via the standard maps (e.g., $(x_i) \mapsto (x_i x_j)_{i \leq j}$ for projective-space vectors $(\cdots x_j \cdots)$) would come at a price of quadratically increasing the lowest possible ambient dimension Simanca (2018); permutation_matrix (https://math.stackexchange.com/users/913340/permutation matrix); reuns (https://math.stackexchange.com/users/276986/reuns). We thus stick with our original mapping in order to be able to run smaller-scale numerical experiments.

### STIEFEL MANIFOLD REVISITED: THE "ST (VEC)" EMBEDDING

A point in the Stiefel manifold can be represented by a point on the Grassmannian and a matrix $\mathcal{V} \in \mathrm{SO}(k)$. In other words, any element of $\mathrm{O}(n)/\mathrm{O}(n-k)$ can be expressed as an element of $\mathrm{O}(n)/\mathrm{O}(n-k) \times \mathrm{SO}(k)$ and an element of $\mathrm{SO}(k)$. Therefore, given an element $(\mathcal{O}, \mathcal{V})$ on $\mathrm{St}(k, \mathbb{R}^n)$, where $\mathcal{O} \in \mathrm{O}(n)$ represents a point on $\mathrm{Gr}^\star(k, \mathbb{R}^n)$ and $\mathcal{V} \in \mathrm{SO}(k)$, we can define an embedding

$$|\tilde{\psi}_{\mathrm{St}}\rangle = |\psi_{\mathrm{Gr}}(\mathcal{O})\rangle \oplus \vec{\mathcal{V}}, \tag{16}$$

recalling that $\vec{\mathcal{V}}$ is the vectorized matrix $\mathcal{V}$. This embedding has dimension $d + k^2$, where $d \geq \binom{n}{k}$, and we denote this by "St (Vec)". This is much less than the dimension of the embedding $\psi_{\mathrm{St}}$, which is $n^k$.

To generate random points on the Stiefel manifold with this embedding, we can just generate a random $\mathcal{O} \in \mathrm{O}(n)$ and a random $\mathcal{V} \in \mathrm{SO}(k)$ and then construct the embedding.

### THE "GR (PROJ)" EMBEDDING

Recall that $\mathrm{Gr}(k, \mathbb{R}^n) = \mathrm{O}(n)/\mathrm{O}(n-k) \times \mathrm{O}(k)$ is the manifold of $k$-dimensional subspaces of $\mathbb{R}^n$. Thus, we can uniquely represent a point on this manifold by a projector that projects onto this corresponding subspace. In particular, given an $n \times k$ orthogonal matrix $\mathcal{O}, \mathcal{O}^T \mathcal{O} = \mathbb{I}_{k \times k}$ representing a point on $\mathrm{St}(k, \mathbb{R}^n)$, we can create the projector $P_\mathcal{O} = \mathcal{O}\mathcal{O}^T$ that projects onto the span of the columns of $\mathcal{O}$. From this projector, we define the "Gr (Proj)" embedding as its vectorization $\vec{P}_\mathcal{O}$.

As is, the Gr (Vec) and Gr (Proj) embed different spaces — $\mathrm{O}(n)/\mathrm{O}(n-k) \times \mathrm{SO}(k)$ versus $\mathrm{O}(n)/\mathrm{O}(n-k) \times \mathrm{O}(k)$ — but the intrinsic dimension remains the same.

### FLAG MANIFOLDS AND THE "FLAG (VEC)" EMBEDDING

In this section, we consider a general flag manifold $\frac{\mathrm{O}(n)}{\mathrm{SO}(k_1) \times \cdots \times \mathrm{SO}(k_t) \times \mathrm{O}(n-k)}$ where $\sum_{i=1}^t k_i = k$. We note, as with the Grassmannians, that our definition of the Flag manifolds also slightly differs from the standard. Namely, the typical Flag manifold is $\frac{\mathrm{O}(n)}{\mathrm{O}(k_1) \times \cdots \times \mathrm{O}(k_t) \times \mathrm{O}(n-k)}$. We will add these to QuIIEst in future work.

We begin with the case $t = 2$, as presented in Table 3. Again, in the notation of Section D.1, we work with the representation $\Lambda: \mathcal{O} \mapsto \mathcal{O}^k$, and we use the fiducial vector

$$|\mathcal{H}\rangle = \left( \frac{1}{\sqrt{k_1!}} \sum_{\sigma \in S_{k_1}} |\sigma(1)\rangle \otimes \cdots \otimes |\sigma(k_1)\rangle \right) \otimes \left( \frac{1}{\sqrt{k_2!}} \sum_{\sigma \in S_{k_2}} |\sigma(k_1 + 1)\rangle \otimes \cdots \otimes |\sigma(k_1 + k_2)\rangle \right).$$
$$\tag{17}$$

This yields the embedding

$$|\psi_F(\mathcal{O})\rangle = \Lambda(\mathcal{O}) |\mathcal{H}\rangle \tag{18}$$

$$= \frac{1}{\sqrt{k_1! k_2!}} \sum_{j_1,\dots,j_k=1}^{n} \det\!\big(\mathcal{O}_{(j_1,\dots,j_{k_1}),(1,\dots,k_1)}\big) \det\!\big(\mathcal{O}_{(j_{k_1+1},\dots,j_k),(k_1+1,\dots,k)}\big)|j_1\rangle\otimes\cdots\otimes|j_k\rangle \tag{19}$$

$$= \frac{1}{\sqrt{k_1! k_2!}} \sum_{\substack{Q\subset\{1,\dots,n\}\\|Q|=k_1}} \sum_{\substack{P\subset\{1,\dots,n\}\\|P|=k_2}} \sum_{\sigma\in S_{k_1}} \sum_{\pi\in S_{k_2}} \times \tag{20}$$

$$\times \operatorname{sgn}(\sigma)\operatorname{sgn}(\pi)\det\!\big(\mathcal{O}_{Q,(1,\dots,k_1)}\big)\det\!\big(\mathcal{O}_{P,(k_1+1,\dots,k)}\big)|\sigma(Q_1)\rangle\otimes\cdots\otimes|\sigma(Q_{k_1})\rangle\otimes|\pi(P_1)\rangle\otimes\cdots\otimes|\pi(P_{k_2})\rangle$$

$$= \sum_{\substack{Q\subset\{1,\dots,n\}\\|Q|=k_1}} \sum_{\substack{P\subset\{1,\dots,n\}\\|P|=k_2}} \det\!\big(\mathcal{O}_{Q,\{1,\dots,k_1\}}\big)\det\!\big(\mathcal{O}_{P,\{k_1+1,\dots,k\}}\big)|Q\rangle\otimes|P\rangle . \tag{21}$$

As is, this embedding is into $\mathbb{R}^{n^k}$, but because an orthonormal basis is given by tensor products of $|Q\rangle$, we see that this embedding works into $\mathbb{R}^d$ for any $d \geq \binom{n}{k_1}\binom{n}{k_2}$. We call this the "Flag (Vec)" embedding.

A straightforward extension to general $t$ yields an embedding of $\frac{\mathrm{O}(n)}{\mathrm{SO}(k_1)\times\cdots\times\mathrm{SO}(k_t)\times\mathrm{O}(n-k)}$ into $\mathbb{R}^d$ for any ambient dimension $d \geq \prod_{i=1}^{t}\binom{n}{k_i}$.

THE "PAULI" EMBEDDING

We would like to construct an embedding for the quotient space $\mathcal{G}/\mathcal{H} = \mathrm{U}(n)/\mathcal{P}_n$, where $\mathcal{P}_n = \langle e^{i\frac{2\pi}{n}}, X, Z\rangle$ is the Pauli (a.k.a. Heisenberg-Weyl) group of prime dimension $n$, and where $Z$ and the real-valued $X$ are the standard $n$-dimensional qudit Pauli matrices Bittel et al. (2025). To construct the quotient space using the Gilmore-Perelomov prescription in Section D.1, we require a vector $|\mathcal{H}\rangle$ in some representation of $\mathrm{U}(n)$ that is invariant under $\mathcal{P}_n$ and not invariant under any $\mathrm{U}(n)$-subgroup that contains $\mathcal{P}_n$.

We pick the $n^4$-dimensional four-fold tensor-product unitary representation $\Lambda\colon U \mapsto U\otimes\overline{U}\otimes U\otimes\overline{U}$ for $\mathrm{U}(n)$. The corresponding Pauli representation is then $Z\otimes\overline{Z}\otimes Z\otimes\overline{Z}$ and $X\otimes X\otimes X\otimes X$, where we recall that $X$ is real. There is an $n^2$-dimensional subspace $S$ that is invariant under this Pauli representation. It is spanned by the vectors

$$|\overline{a,b}\rangle = \frac{1}{\sqrt{n}} \sum_{c\in\mathbb{Z}_n} |c, c+a, c+b, c+a+b\rangle , \tag{22}$$

where $a, b \in \mathbb{Z}_n$, and where addition inside the kets is done modulo $n$. Our "Pauli" embedding is constructed by defining $|\mathcal{H}\rangle$ to be a random unit vector in $S$. Then, as in Section D.1, we apply a the unitary rotation in the four-fold tensor-product representation, and embed the resulting $n^4$-dimensional complex vector into $\mathbb{R}^{2n^4}$.

We now narrow down the quotient space that is spanned by our "Pauli" embedding. It has also been shown Ref. Bittel et al. (2025) that $S$ is not invariant under the larger Clifford group $\mathcal{C}_n \supseteq \mathcal{P}_n$, defined as the normalizer of the Pauli group inside the unitary group. Leaving open the possibility that there exists some "in-between" group $\mathcal{P}_n^\star$ satisfying

$$\mathcal{P}_n \subseteq \mathcal{P}_n^\star \subset \mathcal{C}_n , \tag{23}$$

we conclude that the quotient space of the "Pauli" embedding is $\mathrm{U}(n)/\mathcal{P}_n^\star$. Since the Clifford group is finite, the intrinsic dimensions of both $\mathrm{U}(n)/\mathcal{P}_n^\star$ and $\mathrm{U}(n)/\mathcal{P}_n$ are equal to the dimension $n^2$ of the unitary group.

## D.3 EXTENSION TO DOUBLE COSET SPACES

Note: this section has been added during rebuttal

The coherent-state method, introduced above, produces vectors $|\psi_{\mathcal{G}/\mathcal{H}}(a)\rangle$ valued in a homogeneous space $\mathcal{M} = \mathcal{G}/\mathcal{H}$ for coset representatives $a$. Such vectors transform under elements $g \in \mathcal{G}$ according to the group's multiplication rules, $|\psi_{\mathcal{G}/\mathcal{H}}(a)\rangle \to |\psi_{\mathcal{G}/\mathcal{H}}(ga)\rangle$.

A double coset space $\mathcal{K}\backslash\mathcal{G}/\mathcal{H}$ can be constructed from these vectors by taking their orbits under the subgroup $\mathcal{K}$,

$$|\mathcal{K}a\mathcal{H}\rangle = \frac{1}{\sqrt{|\mathcal{K}|}} \sum_{k \in \mathcal{K}} |\psi_{\mathcal{G}/\mathcal{H}}(ka)\rangle \tag{24}$$

for all coset representatives $a$. Above, we use a sum to denote the linear combination of vectors, but this can substituted for an integral over the Haar measure in the case of a continuous group.

As an example, let us take the case of the sphere, $\mathcal{M} = \mathrm{SO}(3)/\mathrm{SO}(2)$, and consider $\mathcal{K} = \mathrm{SO}(2)$ to be the group of proper rotations around the **z**-axis. The orbit of a generic point on the sphere under this subgroup is the latitude on which the point is located. Therefore, double cosets are labeled by points in a fixed longitude connecting the north and south poles, which is no longer a manifold but an orbifold Albert et al. (2020). The reason is because the poles do not move under $\mathcal{K}$-rotations and are singular points whose neighborhood does not resemble flat space.

Double-coset spaces for finite subgroups of $\mathrm{SO}(2)$ come with enlarged versions of a longitudinal line that are called spherical lunes. In such cases, the poles remain the two singular points of these orbifolds. In general, orbifolds of other discrete subgroups of $\mathrm{SO}(3)$ correspond to fundamental domains — patches of the sphere whose translates under $\mathcal{K}$ tile the entire sphere.

# E   OVERVIEW OF IDE METHODS TESTED

**lPCA :**   Since a manifold is locally isomorphic to $\mathbb{R}^{d_i}$, the directions normal to the hyperplanes have zero variance. Given $k$ nearest-neighbor (NN) of a point sampled from $\mathbb{R}^D$, singular value decomposition (SVD) of the $k \times D$ matrix is performed to determine the principal components. The intrinsic dimension of the manifold is then estimated by different means:

- 'maxgap' : the component showing the biggest spectral gap is returned as the intrinsic dimension, i.e.

$$\hat{d}_i(x; k) = \mathrm{argmax}_{j \in 1, \ldots, \min(D,k)-1} \frac{e_{j-1}}{e_j} \tag{25}$$

- 'ratio' : the minimum number of components needed to explain $1 - \epsilon$ of the total variance.

$$\hat{d}_i(x; k, \epsilon) = \min\{j : R_j \geq 1 - \epsilon\} \tag{26}$$

  where $R_j := \frac{\sum_{i=1}^{j} \sigma_{[i]}^2}{\sum_{i=1}^{N} \sigma_{[i]}^2}$ is the cumulative variance ratio $((.)_{[i]}$ denotes that the quantity is sorted from largest to smallest (algebriacally).

- 'fo' : the index $j$ for which the (sorted) eigenvalues cross $(1 - \epsilon)$ of the largest eigenvalue.

$$\hat{d}_i(x; k, \epsilon) = \min\{j : \sigma_{[i]}^2 \geq (1 - \epsilon)\sigma_{[k]}^2\} \tag{27}$$

It is easy to check that these definitions are equivalent with suitable choices of the hyperparameter $\epsilon$ for manifolds with a *clear* spectral gap. In order to preserve computational resources, we therefore report the results of the 'maxgap' technique in the main section, but we also report results of small-scale experiments with the 'ratio' and 'fo' versions in G.1 We created a custom function to compute the $\hat{d}_i$ according to Eq. 25 above, closely drawing from the implementation in scikit-dimension.

The following methods, namely MLE, DANCo, CorrInt and TwoNN were implemented by accessing the implementations directly from the `scikit-dimension` package.

**MLE :**   Given a set of samples $X_1, X_2, \ldots X_n$, related to a sample in lower-dimensional space $m$ and equipped with a smooth density $f$, one considers the Possion process $\lambda(t)$ of number of points in a small sphere around the point with radius $t$. The log-likelihood of this process (assuming a constant density $f(x)$ in the sphere of radius $R \geq t$) yields

$$\hat{d}_i(x; R) = [\frac{1}{N_R(x)} \sum_{j=1}^{N_R(x)} \log \frac{R}{T_j(x)}]^{-1} \tag{28}$$

where $T_j(x)$ is the Euclidean distance between $x$ and its $j$-th NN. Some results on the statistics of these log distances were considered too.

Haro et al. (2008) goes one step further and considers a noisy translation of the observed distances, which leads to a non-linear recursive equation, which they can solve self-consistently. In particular for isotropic Gaussian noise with scale $\sigma$, their ID estimate reads

$$\hat{d}_i(x; R) = \Big[ \frac{1}{N_R(x)} \sum_{j=1}^{N_R(x)} \frac{\int_{r=0}^R \exp(-\frac{(T_i-r)^2}{2\sigma^2}) \log(\frac{T_k}{r}) dr}{\int_{r=0}^R \exp(-\frac{(T_i-r)^2}{2\sigma^2}) dr} \Big]^{-1} \tag{29}$$

**CorrInt :** For a set $X_1, ..., X_n$ of i.i.d. samples with a smooth density $f(x)$ in $\mathbb{R}^{d_i}$, the Euclidean distance between a point $x$ and its $k$-th NN $T_k(x)$ satisfies

$$k/n \approx f(x) V_{d_i}(T_k(x)) \tag{30}$$

where $V_{d_i}(R)$ is the volume of a $d_i$-dimensional sphere of radius $R$. Since $V_{d_i}(R) \propto R^{d_i}$, the number of points $k$ in a ball of radius $R$ grows exponentially with $d_i$. This is the motivation behind the fractal dimension definition.

One computes the correlation integtral (sum) as

$$C(r) = \frac{1}{N(N-1)} \sum_{i>j} \mathbf{1}_{||\mathbf{x}_i - \mathbf{x}_j|| \leq r} \tag{31}$$

One can then estimate the dimension as

$$\hat{d}_i(x; k_1, k_2) = \frac{\log(C(r_2)/C(r_1))}{\log(r_2/r_1)} \tag{32}$$

where the hyper-parameters $k_1$ and $k_2$ are used to find the median distances $r_1$ and $r_2$

**TwoNN :** For a constant density $\rho$ around a point $x$, the volume of the hyper-spherical shell between $i$ and $i+1$-th NN is drawn from an exponential distribution in the volume $\Delta\nu_l = \omega_{d_i}(r_l^{d_i} - r_{l-1}^{d_i})$. Define $R = \frac{\Delta\nu_2}{\Delta\nu_1}$ and then it follows that $f(R) = (1+R)^{-2}$. It then follows that $f(\mu) = d_i \mu^{-d_i-1}$ where $\mu = \frac{r_2}{r_1} \in [1, \inf)$. (Basically note that $f(\mu) = f(R)\frac{dR}{d\mu}$ and $R = \mu^{d_i} - 1$) To make matters less prone to computational errors, discard $\alpha$ of the largest $\frac{r_2}{r_1}$ values. Thus the ID estimate is given by

$$\hat{d}_i(x; k, \alpha) = -\frac{\log(1 - F(\mu))}{\log \mu} \tag{33}$$

where $\mu$ represents the ratio $\frac{r_2}{r_1}$ for the $k$NNs of the particular point.

**DANCo :** For a manifold $\mathcal{M} \subseteq \mathbb{R}^{d_i}$, consider an embedding $\phi : \mathbb{R}^{d_i} \to \mathbb{R}^D$ which is locally isometric, smooth and possibly non-linear. Then the points in a local neighborhood are drawn uniformly from the hyperspheres. The distribution for distances for the unit hypersphere normalized by the distance of the $k$-th NN follows the distribution

$$g(r; k, d_i) = k d_i r^{d_i-1} (1 - r^{d_i})^{k-1}$$

while the mutual angles follow the von Mises-Fisher (VMF) distribution

$$q(\mathbf{x}; \nu, \tau) = \mathbf{C_{d_i}}(\tau) \exp(\tau \nu^{\mathbf{T}} \mathbf{x})$$

where $C_{d_i}(\tau)$ is a normalization constant. It should be noted that the parameter $\tau$ in the VMF distribution denotes the concentration of angles around the mean — the parameter $\tau = 0$ reducing this distribution to the uniform distribution on the sphere. The joint distribution of the normalized distance and mutual angles factorizes into the product of marginals for the unit hypersphere. The ID

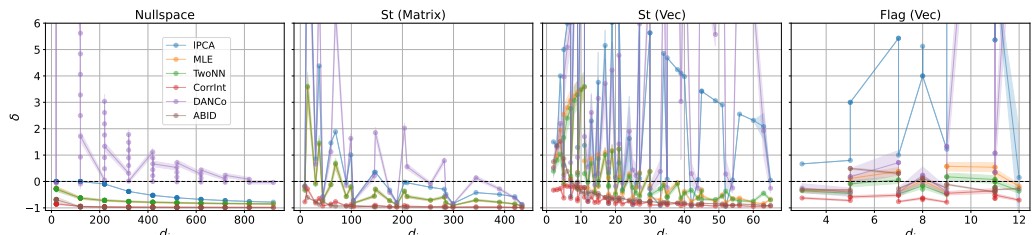

Figure 12: Scaling of relative error with ID for the other manifolds. We observe the same sort of trend as discussed in the main text, except for flag, which has limited resolution.

is then estimated by minimizing the KL-divergence between the theoretical and experimental joint distribution of normalized distance and mutual angles.

$$\hat{d}_i(x;k) = \text{argmin}_{d=1,\ldots,d_a} \int_{-\pi}^{\pi} d\theta \int_0^1 dr h_d(r,\theta) \log(\frac{\hat{h}_d(r,\theta)}{h_d(r,\theta)}) \tag{34}$$

where $\hat{h}$ refers to the experimental joint distribution, and $h(r,\theta) = g(r) \cdot q(\theta)$ is the theoretical joint distribution.

**ABID :** As discussed in Thordsen & Schubert (2022), the distribution of pairwise cosines between two points drawn randomly and uniformly from a $d$-ball (excluding the origin; also holds for any such spherical distribution) follows a Beta distribution on the interval [-1,1].

$$P(\cos\theta) = \frac{1}{2}B(\frac{1+\cos\theta}{2}; \frac{d_i-1}{2}, \frac{d_i-1}{2})$$

from which it follows that

$$\mathbb{E}[\cos^2\theta] = d_i^{-1}$$

This motivates the following definition for the ABID ID estimator

$$\hat{d}_i(x;k) = (\mathbb{E}_{\mathbf{x}_i,\mathbf{x}_j \sim B_k(x)}[\cos^2(\mathbf{x}_i,\mathbf{x}_j)])^{-1} \tag{35}$$

where $B_k(x)$ denotes the ball containing the kNN of x, i.e. a ball of radius $T_k(x)$

## F    COMPARISON WITH OTHER BENCHMARKS

Here we present results of comparing our manifolds against other standard manifolds. In Appx. F, we present evidence that our manifolds are also adversarial as compared to other standard benchmarks. Note that one of our main considerations, was to compare these manifolds at fixed parameters and resources, hence we only include manifolds for which we can tune the $d_i$ and $d_a$. This is the reason why we exclude some manifolds like the Moebius strip, torus, Swiss rolls, etc.

**Isotropic Gaussian vectors**    Based on Hein & Audibert (2005), we compare our manifold against isotropic Gaussian vectors $\in \mathbb{R}^{d_i}$ linearly embedded into $\mathbb{R}^{d_a}$, cf. Fig'7

**Affine spaces**    Affine spaces are isomorphic to the linear nullspaces we considered in our main text. Given $d_i, d_a$, the nullspace of matrix $A \in \mathbb{R}^{d_a \times d_a}$ with rank $d_a - d_i$ consists of a hyperplane of intrinsic dimension $d_i$. In order to preserve the "smoothness" of our manifold, we sample the coefficients of the basis vectors of the nullspace from the standard unit normal, cf. Fig'8

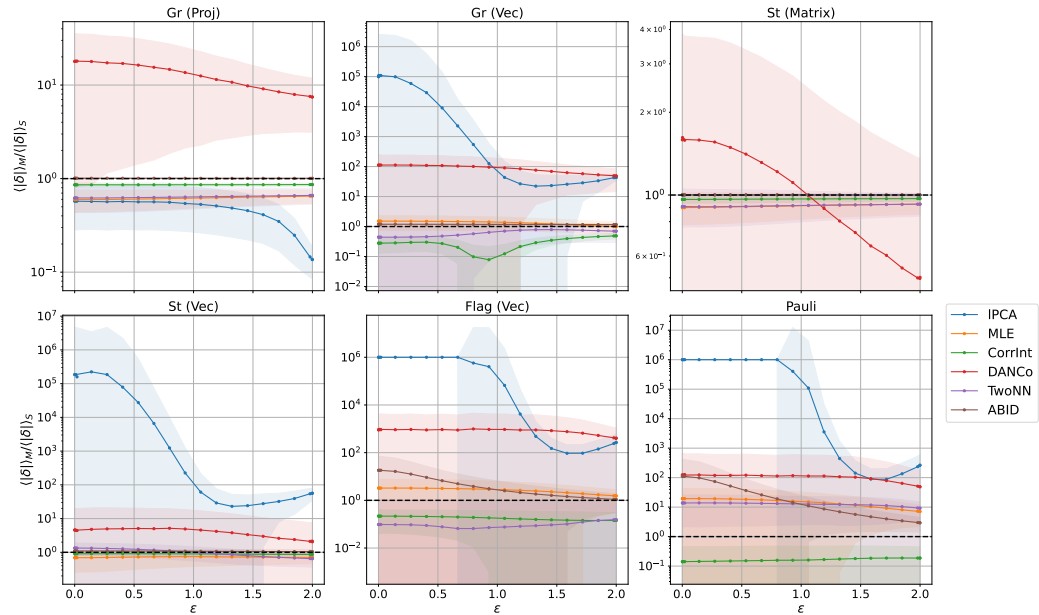

Figure 13: Comparison of errors when distorting QuIIEst manifolds versus spheres. The large change indicates that spheres become drastically more difficult to do IDE on with increasing distortion, as opposed to QuIIEst manifolds.

**Nonlinear manifolds** Based on Rozza et al. (2012); Campadelli et al. (2015), we use a generalized version of the manifolds denoted by $\mathcal{M}_\beta$ — where we uniformly sample from $X : [0, 1)^{d_i}$ and then construct $Y = \sin(\cos(2\pi X))$, and finally linearly embed it into $\mathbb{R}^{d_a}$. This differs from the original formulation in that they append a $\tilde{Y} = \cos(\sin(2\pi X)$ and finally duplicate this to get $d_a = 4d_i$, cf. Fig '9

For the purposes of this experiment, we run extensive scaling experiments by choosing small-dimensional manifolds. In particular we choose Grassmannians with ID of 2,3,4,5; Stiefels with ID of 3,5; Flags with ID of 4,12 and Pauli with ID of 3.

# G ADDITIONAL EXPERIMENTS AND PLOTS

## G.1 OTHER VERSIONS OF LOCAL PCA

We observe that the $\hat{d}_i$ from lPCA vary significantly on the hyperparameter $\epsilon$; in particular this seems to suggest that there is no clear spectral gap in the singular values of the data covariance matrix $XX^T$. However, we do note that there always exist some value of $\epsilon^*$ for which $\delta(\epsilon^*) = 0$, but there is no clear pattern or consistent value for the choice of $\epsilon^*$ even for distinct versions for the same manifold, atleast within the scope of the experiments performed. We thus relegate this interesting investigation to a future project. The results are plotted in Fig. 10.

## G.2 SQUEEZING : COMPARISON WITH SPHERES

We present in Fig. 17a the effect of distortion for spheres, as compared to QuIIEst manifolds. In particular, we observe that spheres exhibit a very large increase in the error rate as anisotropy is increased. In conjunction with Section 5.3, this demonstrates that QuIIEst manifolds are already challenging enough for the IDE methods tested, and enhances it applicability as a more robust performance evaluator for IDE.

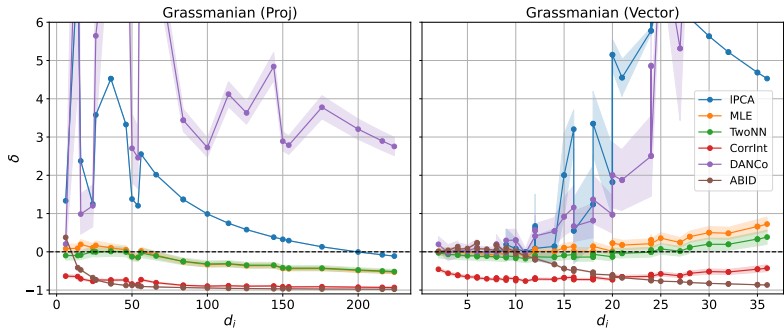

Figure 14: Effect of scaling with intrinsic dimension within the same family of manifolds. The relative error $\delta$ is plotted as a function of increasing $d_1$. Most manifolds show a transition from overestimation at small $d_i$ to underestimation at high $d_i$, corroborating earlier observations for other manifolds Levina & Bickel (2004). The Gr (Vec) embedding shows some minor differences at the same range for $d_i$, but is overall consistent.

### G.3 SCALING EXPERIMENT PLOTS

#### G.3.1 SCALING WITH INTRINSIC DIMENSION

We include plots for scaling intrinsic dimension within the same manifold. In partricular, we plot the different embeddings for the Grassmannian manifolds, where we observe a transition from over-estimation of the ID to underestimation at some point, indicating an optimal ID where the methods perform the best. This corroborates previous experiments at fixed sample size and hyper-parameter search.

We include here some other plots on scaling of the relative error for different manifolds in Fig 12. We once again notice the trend mentioned in the main text, where an initial over-estimation gives way to an under-estimation. This trend is not immediately obvious for the Flag manifolds due to lack of sufficient points. The Pauli manifold is excluded since we could only test 4 different IDs.

#### G.3.2 SCALING WITH SYSTEM SIZE

The dependence of the intrinsic dimension estimate $\hat{d}_i$ comes from the fact that the fraction of neighbors in a small neighborhood for manifolds are given by Levina & Bickel (2004)

$$\frac{k}{N} = \Omega_{d_i}\rho(x)T_k^{d_i} \tag{36}$$

Thus in order to get a uniform and locally dense sampling of points, we need to hold $k/(T_k^{d_i}N)$ fixed, which translates to the condition that

$$N \propto \exp(-d_i)$$

We investigate the effect of changing $N$ while holding hyper-parameters fixed. We logarithmically sample $N$ so that $N/d_i$ goes from 2 to 300. The results are summarized in Fig 15

### G.4 CORRELATION OF IDE PERFORMANCE WITH STATISTICAL PROPERTIES

We present the results for the correlation of IDE performance with VDI, Tr $\mathbf{\Sigma}$ and $\langle R^2 \rangle$ here. Most methods seem to show no dependence on the total variance, with few exceptions being DANCo for linear nullspaces. On the other hand, there is a weak positive correlation between performance and inter-component correlation. The latter makes *a posteriori* sense since this implies that the manifold embeddings show structrual similarites, which makes it easier for the methods to discover the latent dimension.

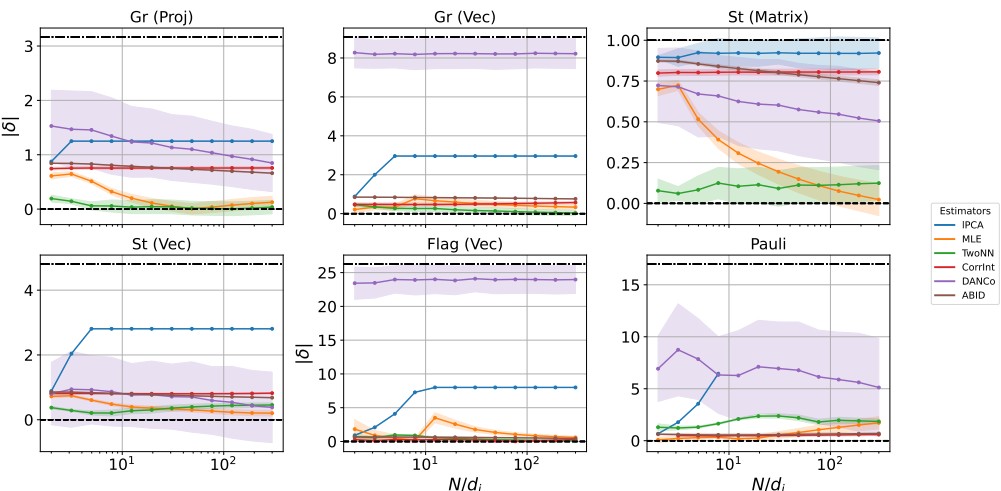

Figure 15: Performance of IDE methods as a function of $N/d_i$. We observe a gradual convergence with increasing sample size. The shaded region shows the 1-$\sigma$ error in the local ID estimates, averaged over three seeds.

### G.5 Geometric Properties

We measure the local mean curvature $H(x)$ and the local density $\rho(x)$. From these two measurements, we calculate a dimensionless parameter $\kappa(x) = \rho/H^{d_i}$. In order to find the curvature, we fit a quadratic surface in a local neighborhood and use standard results due to Gauss. We sample a kNN-neighborhood and estimate the density based on the ratio $k/\Omega_{d_i} T_k^{d_i}$. We sample 5 logarithmically-spaced $k$ values and choose the median value as the desired geometric property. The tabulated results are then shown in Table 5

| Quantity | Gr (Vec) | Gr (Proj) | St (Matrix) | Pauli | St (Vec) | Flag (Vec) |
|---|---|---|---|---|---|---|
| $\langle H \rangle$ | 1.0313 | 1.1417 | 0.6702 | 0.4667 | 0.5348 | 1.5032 |
| $\langle \rho \rangle$ | 0.8527 | 0.5933 | 0.1766 | 0.0373 | 0.2846 | 2.9473 |
| $\langle \kappa \equiv \rho/H^{d_i} \rangle$ | 0.7758 | 0.3736 | 0.9097 | 0.8430 | 2.1033(*) | 0.5772 |
| $\langle |\delta| \rangle$ | **0.2023** | 0.5785 | 0.5793 | 0.5846 | 0.8502 | 2.4085 |

Table 5: Average absolute values by manifold (manifolds sorted by increasing $\langle |\delta| \rangle$).
(*) This number was skewed by a manifold with intrinsic dimension $d_i = 12$ which we do not include in the table. Including that we get 4080.9902.

### G.6 Additional plots on distortion experiments on MNIST

We provide additional plots here regarding our distortion experiments in MNIST. As emphasized in the main text, our observations are similar qualitatively to the analogous experiments for QuIIEst manifolds.

### G.7 Experiments with other statistical averages

We hereby report the result of replacing the mean as the GID estimate with four other statistical quantities — the median, the mode, the median of means and the mean of medians. The results are summarized in Fig.18. The relative squared errors between mean and the other quantities are respectively are $1.9 \times 10^{-3}, 9.0 \times 10^{-2}, 5.2 \times 10^{-5}, 7.9 \times 10^{-4}$, suggesting that the different statistical estimates are consistent with each other.

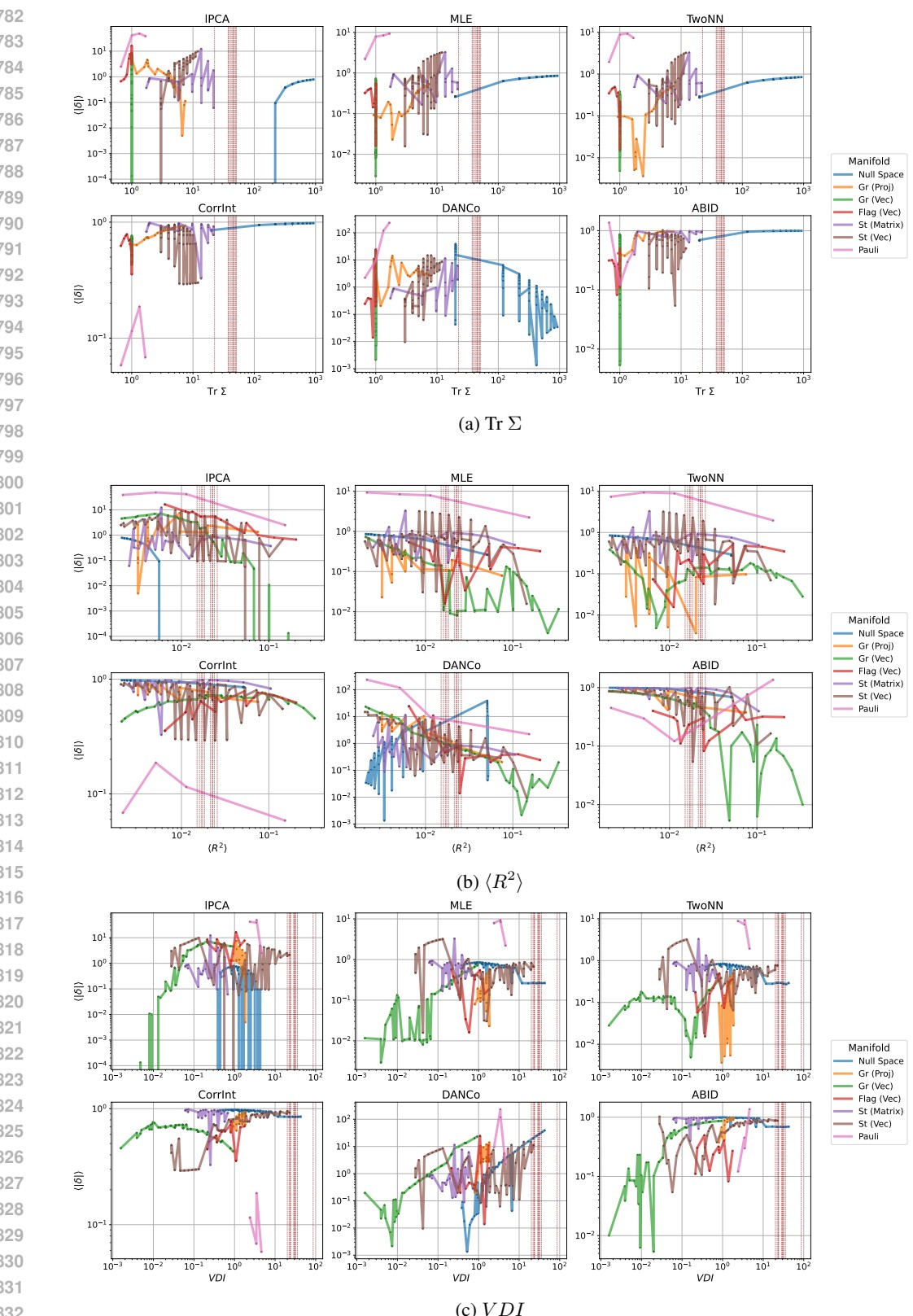

(a) Tr $\Sigma$

(b) $\langle R^2 \rangle$

(c) $VDI$

Figure 16: IDE performance seems to be almost independent for Tr $\Sigma$ with slight dependence observed for the angle-based methods. On the other hand, most methods except lPCA, seem to show weak positive correlation between performance and $\langle R^2 \rangle$. We observe that DANCo and ABID show a prominent negative correlation of performance with anisotropy, with ABID saturating at VDI $\sim 1.0$. The light vertical maroon lines represent the corresponding quantities for different classes in MNIST.

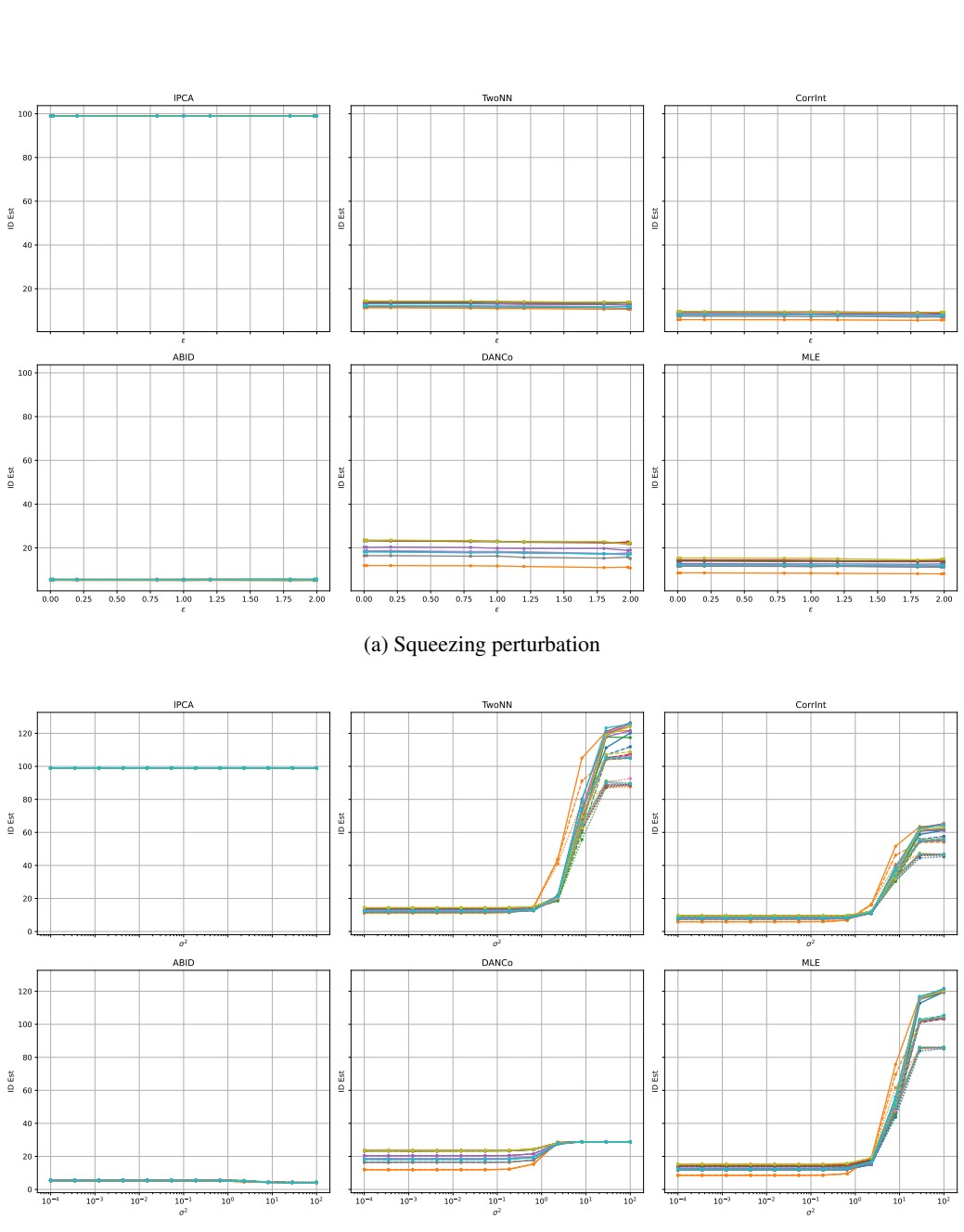

(a) Squeezing perturbation

(b) Additive noise perturbation

Figure 17: Effect of (a) distortion and (b) additive noise perturbation on MNIST dataset. We observe minimal degradation in (a) and degradation when $\sigma^2 \sim \mathcal{O}(1)$ in (b), both observed for QuIIEst manifolds as well.

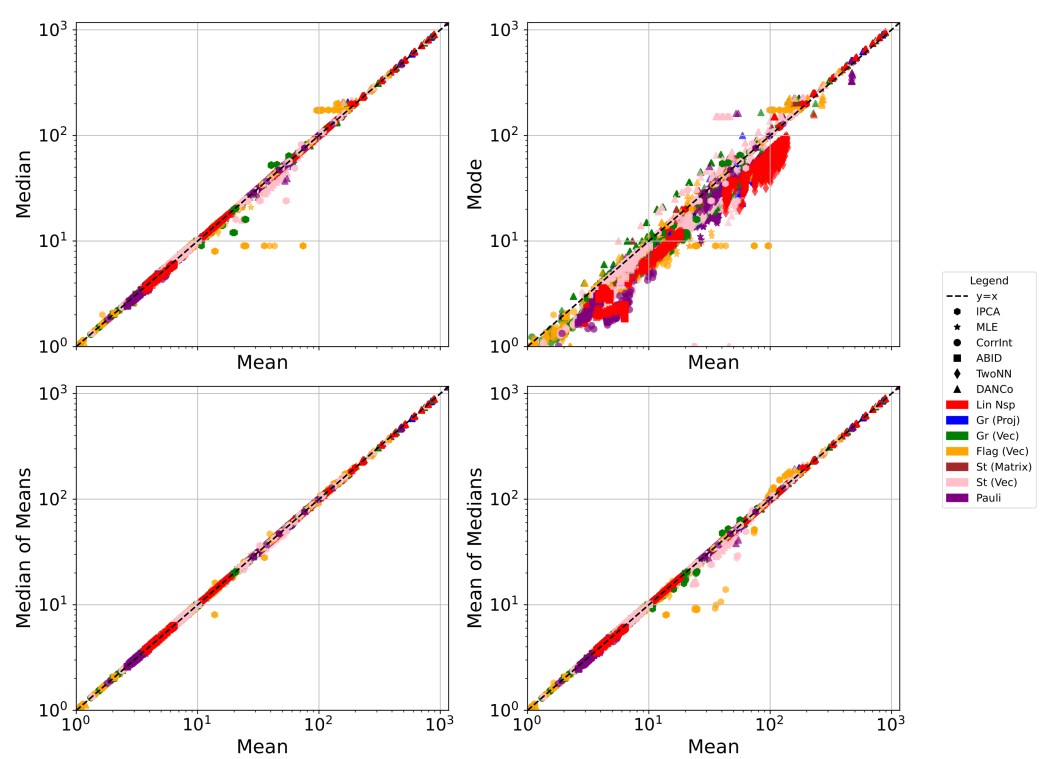

Figure 18: Results of using different statistical quantities. Most of them are concentrated around the $y = x$ line, indicating that they are quite consistent with each other.

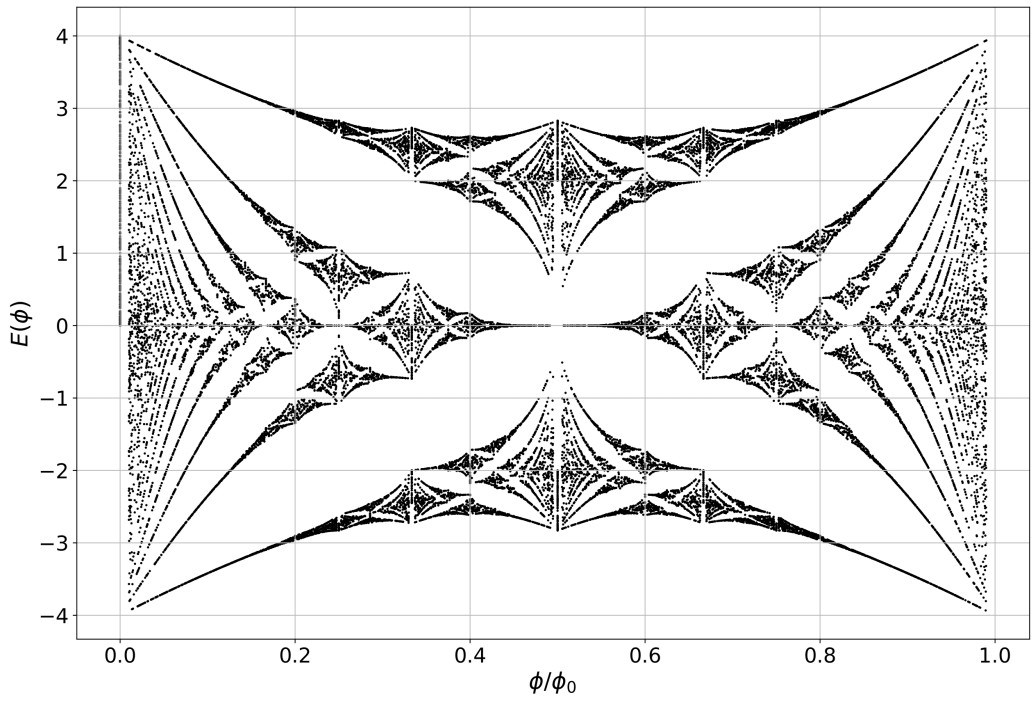

Figure 19: The Hofstadter butterfly. Numerical simulations indicate a fractal dimension of $d_i = 1.445$.

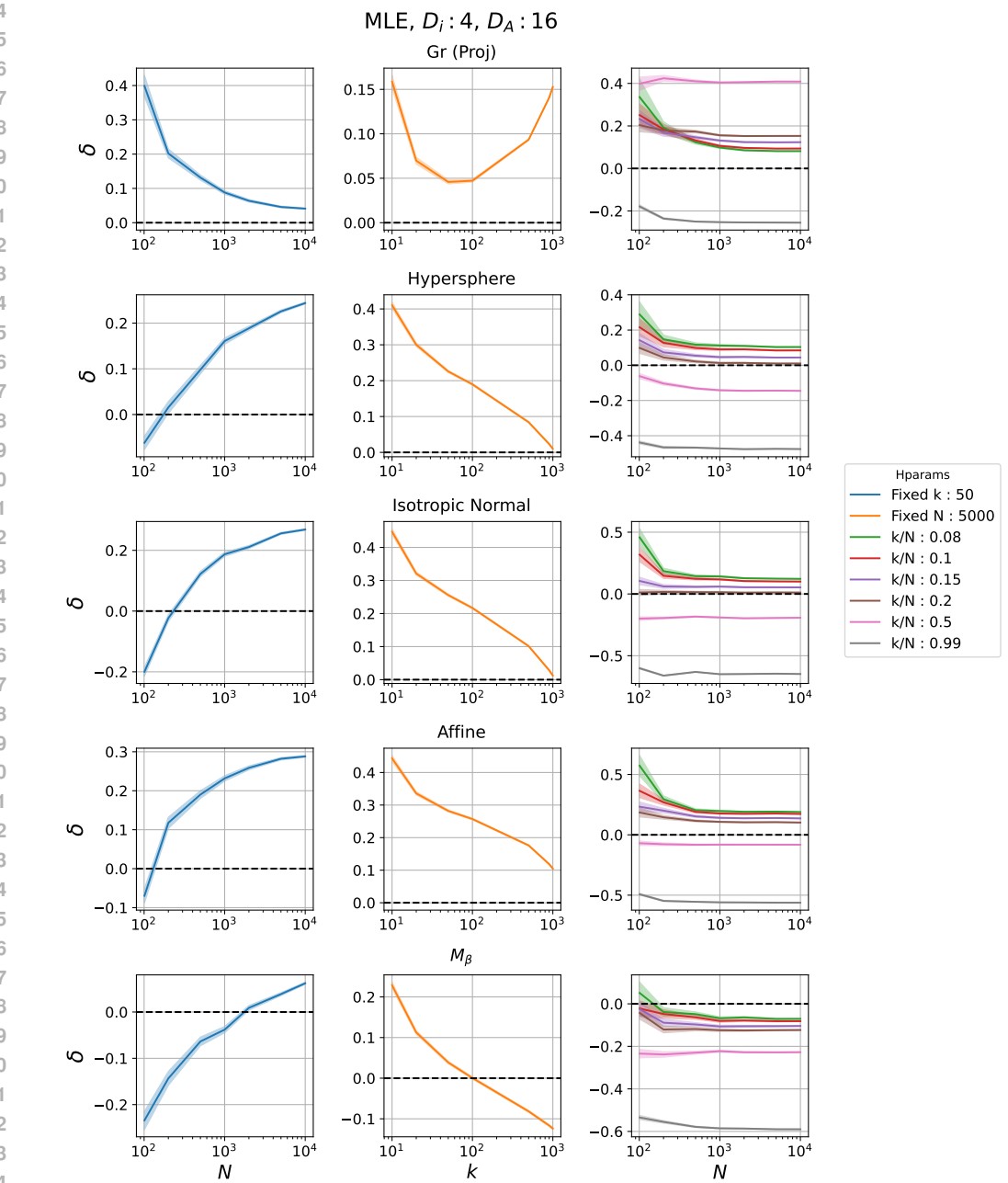

Figure 20: Results of extensive hyper-parameter run with MLE on grassmaian projector representation.

## G.8 EFFECT OF HYPER-PARAMETERS

We present a representative run for hyper-parameter sweep for the MLE method for Grassmannian manifolds. We notice that the absolute error decreases as a function of $N$, but then it switches from underestimation to overestimation. We also see that the error decreases mostly with increasing $k$ when $N$ is held fixed. Holding $k/N$ fixed on the other hand, results in convergence, but **not** necessarily to $\delta \approx 0$.

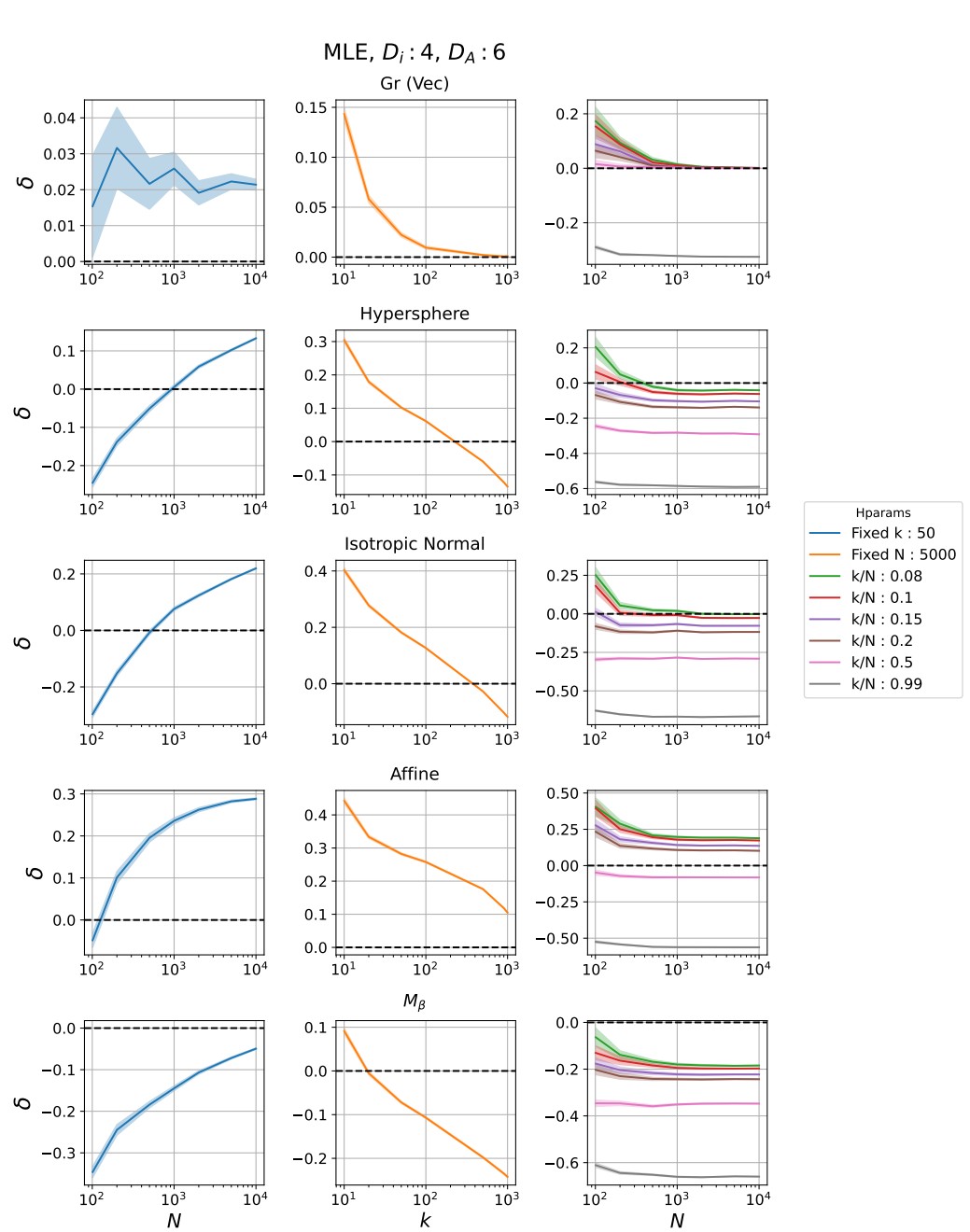

Figure 21: Results of extensive hyper-parameter run with MLE on grassmaian vector embedding.

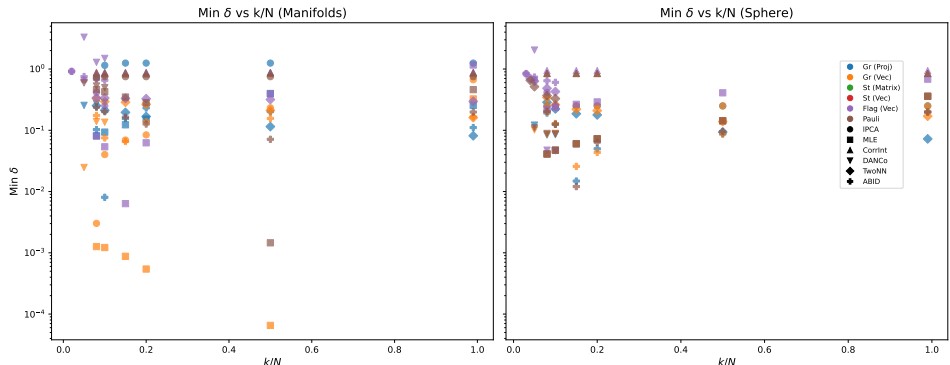

Figure 22: We present a summary of our results of the extensive hyper-parameter scaling performed in Sec. G.8. In particular, we present the results of scaling up both $k$ and $N$ holding $k/N$ constant. As we show here, there is not much difference at this scale. Manifolds typically had $d_i$ around $20 - 30$.

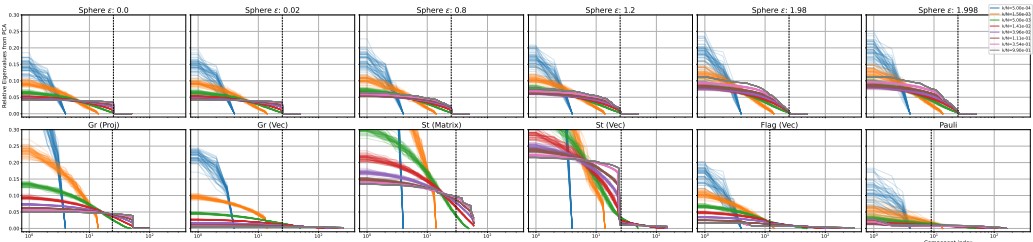

Figure 23: Extra plots for lPCA various manifolds. $\epsilon$ denotes the degree of anisotropy — higher $\epsilon$ indicates a higher anisotropy. Notice that only the undistorted sphere (top left) shows the expected behavior of a sharp drop at the ID.

### G.9 ASYMPTOTIC ANALYSIS

At our scale, we already see that manifolds with $d_i \sim 20 - 30$ show only slight deviations in the asymptotic limits of $k \to \infty, N \to \infty$. The actual limit was $N \sim \mathcal{O}(10^4)$.

### G.10 ADDITIONAL LPCA INTERP PLOTS

We present additional plots on lPCA here in the appendix. As alluded to in the main text, our main conjecture is that the kNN-based IDE methods fail due to deviations of their expected distributions. In the case of lPCA, this is true for distorted spheres and QuIIEst manifolds, cf. Fig. 23.

## H FRACTAL CURVES

Fractal curves present an interesting class of objects since they usually have *fractional dimension*. This is because the dimension for fractal curves is determined by the box-counting method, where if the entire box is partitioned into hypercubes of side $\epsilon$, the number of boxes with non-zero points depends on $\epsilon$ through a power-law decay with dimension

$$N(\epsilon) = N_0(\epsilon/\epsilon_0)^{-d}$$

However, fractal curves are *not* manifolds and are not locally isomorphic to hyperplanes, partly due to the discontinuous nature of the fractal sets.

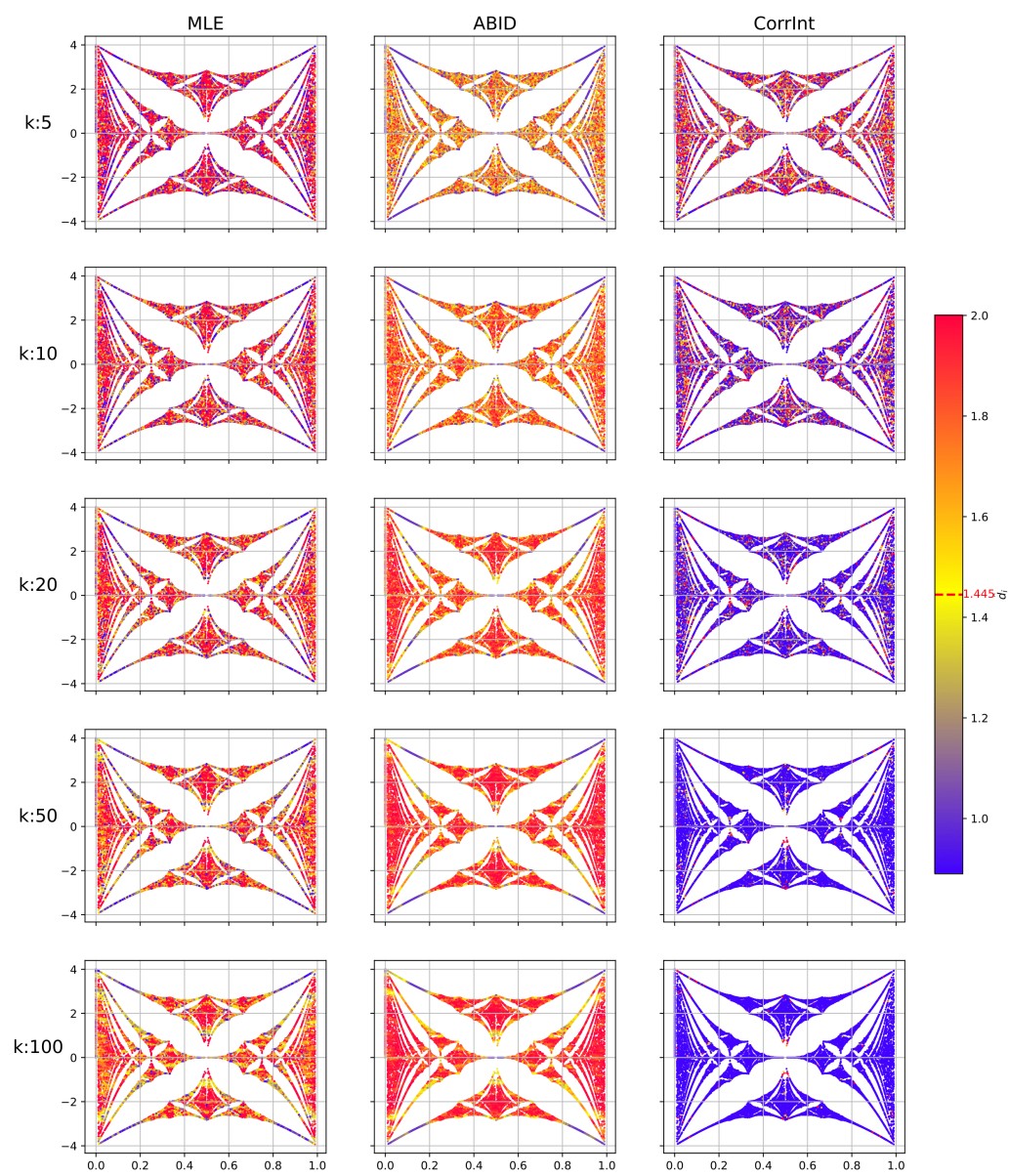

Figure 24: Red dashed line indicates the actual fractal dimension. Row indicates same $k$, column refers to same method. Note how ABID produces ID estimates for small $k$, while most other methods return LID estimates very different from the fractal dimension. $N = 1000$.

## H.1 HOFSTADTER'S BUTTERFLY

Hofstadter's butterfly Hofstadter (1976) is an example of a fractal curve obtained from quantum physics, by solving the system of electrons with nearest-neighbor hopping on a 2D lattice, while being subjected to a perpendicular constant magnetic field. The butterfly emerges when one plots the gapped energy spectra as a function of the magnetic flux through a plauette, as shown in Fig. 19

We include fractals in the appendix because while they are not really manifolds, they can serve as useful tools to test the manifold hypothesis. This is because fractals do have non-uniform local ID, and as such a method to estimate LID *should* give different answers at different points, which can be probed by (1) looking at the standard deviation and (2) visually for the butterfly which lives on a 2D

plane. We report the results of these investigations on the Hofstadter butterfly only with methods which can give fractional ID estimates, in particular, we test MLE, ABID, CorrInt. [2]

By far, ABID seems to perform the best in estimating the ID, with the standard deviation decreasing as $k$ increases, indicating that more and more points are estimated to have the ambient dimension due to the increasing neighborhood size, see Fig. 24 Also the errors fall within the reasonable value of the ambient dimension, thereby showing that ABID can be trusted as a method to estimate if data lies on a *manifold* or not.

| $k$ | MLE | ABID | CorrInt |
|---|---|---|---|
| 5 | $2.285 \pm 2.083$ | $1.485 \pm 0.323$ | $1.858 \pm 1.581$ |
| 10 | $1.808 \pm 1.142$ | $1.620 \pm 0.303$ | $1.397 \pm 1.540$ |
| 20 | $1.663 \pm 0.781$ | $1.690 \pm 0.291$ | $0.885 \pm 0.933$ |
| 50 | $1.625 \pm 0.576$ | $1.742 \pm 0.274$ | $0.598 \pm 0.620$ |
| 100 | $1.631 \pm 0.502$ | $1.765 \pm 0.262$ | $0.486 \pm 0.442$ |

Table 6: Intrinsic dimension estimates (mean $\pm$ std) for different methods and values of $k$. Recall from Figure 24 that the true ID of the butterfly is $d_i = 1.445$.

---

[2]We investigated TwoNN but the native implementation could not handle `NaN` values properly, hence we omit the results here.

