# OpenReview forum: "Quantum-inspired benchmark for Intrinsic Dimension Estimation"
_ICLR.cc/2026/Conference — Submitted to ICLR 2026_

### Official Review · Reviewer_5gjM · 2025-10-31

**Soundness:** 3
**Presentation:** 3
**Contribution:** 3
**Rating:** 6
**Confidence:** 4

**Summary:**

This paper discusses a quantum-inspired algorithm for
intrinsic dimension estimation (IDE) in the context of the "manifold
hypothesis" on the set of data. This is an interesting and important
question.
The idea is to constructs infinite
families of manifolds starting from classical homogeneous spaces
as quotients of semisimple real or complex Lie groups (e.g. spheres,
grassmannians, flags, Stiefel manifolds etc). The authors start from what
they call "quantum optical embeddings" of homogeneous spaces, though they
do not explain and I do not understand the word quantum used in this context.
Starting with the ground truth of ID, from the manifolds (datasets) they
construct, then they proceed to give IDE with their method QuIIEst and
then they benchmark it against more known methods, showing its potential.
They also add some noise and make considerations regarding distortion
and possible topological complexity.

**Strengths:**

As far as I know this idea is new, though I do not understand
the quantum component of it, I think it is worth exploring and can
be of aid in IDE, especially when topology, curvature and more generally
geometry plays a role in data distribution. Moreover the authors
are able to provide a playground for others with "toy datasets" that
may be of help in answering other questions (eg. knowledge transfer).
Their results show performance degradation of existing IDE methods,
validating their QuIIEst model and method.

**Weaknesses:**

There is the surprising fact that the geometry of the manifold
(curvature, density etc) plays no role: actually this should be better
motivated, because it may hide a shortcoming in the theoretical treatment.
Moreover a comparison on this point with other IDE methods is mandatory.

The paper shows that IDE methods perform worse on QuIIEst
but does not deeply analyze why, eg.
which geometric/topological/other features cause errors (link with
previous weakness).

The experiments appear very limited in scope, though there may
be promising as scalability goes.

At some point code and generation scripts must be released for
reproducibility.

**Questions:**

How exactly are the “quantum embeddings” implemented? This
part is not fully clear, it would benefit from more details.

Can QuIIEst generate non-homogeneous manifolds? Can their method
work in reducing data parameters and dimensionality (as for any
homogeneous space a group action can be of help).

Are there known analytic expressions for curvature and metric tensors
in these embeddings and after distorsion? (this is actually an
interesting mathematical question, how geometry changes with noise
addition).

How computationally expensive is manifold generation as ID increases?

---

> ### Author Response · Authors · 2025-11-22
> **Response to Reviewer 5gjM**
>
> We thank the reviewer for their comments. We address them below.
>
> > There is the surprising fact that the geometry of the manifold (curvature, density etc) plays no role: actually this should be better motivated, because it may hide a shortcoming in the theoretical treatment. Moreover a comparison on this point with other IDE methods is mandatory.
>
> After running further experiments, we now postulate that a non-isotropic (i.e., direction-dependent) curvature is an important factor in benchmarking. Further distorting the curvature of either our manifolds or MNIST does not lead to a drop in IDE performance, but distorting the isotropic curvature of a sphere does. This suggests that our manifolds may be accurately simulating real-world data with respect to this particular property.
>
> > The paper shows that IDE methods perform worse on QuIIEst but does not deeply analyze why, eg. which geometric/topological/other features cause errors (link with previous weakness).
>
> We agree with the Reviewer on the lack of robust explainability of our results. In the new version, we perform experiments on lPCA [see Section 6] and observe that, for manifolds with isotropic curvature, the explained variances have a sharp drop at the ID, and the drop becomes smoother once we distort the manifold. As such, the dimension of embedded and undistorted spheres is easily picked up by PCA, as opposed to the dimensions of the other manifolds.
> We also observe a weak correlation between anisotropy and IDE performance for the angle-based methods DANCo and ABID (Fig 16). This makes sense since the anisotropy is a deviation from the expected isotropic (hyper-spherical) distributions assumed for these methods to work well.
>
> > The experiments appear very limited in scope, though there may be promising as scalability goes.
>
> We include asymptotic results for our most extensive hyper-parameter search experiments in the Appendix. We do not expect major qualitative differences with further scaling.
>
> > At some point code and generation scripts must be released for reproducibility.
>
> We have included the code (without documentation) in SupMat for now. Full code release information and plan is provided in Appendix A. Code will be publicly released, with proper documentation, on acceptance.
>
> > Questions:
> >How exactly are the “quantum embeddings” implemented? This part is not fully clear, it would benefit from more details.
>
> We add a non-technical description of our methods (see red text) in Section 3, Homogenous spaces, followed by an explicit calculation for 2-spheres in Appendix D.1.1. We would be glad if the reviewer could let us know if the addition is sufficiently clear for a non-technical audience.
>
> > Can QuIIEst generate non-homogeneous manifolds?
>
> Yes, the coherent state method can be used to generate non-homogenous spaces using double coset spaces. We have discussed this further in Appendix D. In addition, we also consider non-manifolds like fractals, which we highlight in the latest manuscript.
>
> In addition to homogeneous spaces, we also test how IDE performance changes upon distortion of our manifolds under random diagonal general linear transformations in Section 5.3. We observe that performance is not significantly affected by such distortion, indicating that homogeneous spaces seem to be already challenging enough.
>
> > Can their method work in reducing data parameters and dimensionality (as for any homogeneous space a group action can be of help).
>
> We hope to be able to improve IDE by incorporating or estimating curvature information about the manifold. Our controlled setting can provide useful hints for generalization to arbitrary manifolds.
>
> >  Are there known analytic expressions for curvature and metric tensors in these embeddings and after distorsion? (this is actually an interesting mathematical question, how geometry changes with noise addition).
>
> There may be special exactly solvable cases, but we are not aware of succinct analytical expressions for the curvature of all members of each of our manifold families.
>
> > How computationally expensive is manifold generation as ID increases?
>
> As discussed in Fig 1, the main advantage of these manifolds is that their embeddings have an ambient dimension that is polynomial in ID. Thus it is not computationally expensive. In our work, we were bound mainly by memory constraints, which can be improved through bit quantization for example.
>
> We hope our responses have improved your perception of our paper. We will appreciate it if you can increase your score in that case. If not, we are happy to discuss any further concerns.

---

### Official Review · Reviewer_wuJs · 2025-10-31

**Soundness:** 4
**Presentation:** 4
**Contribution:** 2
**Rating:** 6
**Confidence:** 3

**Summary:**

This work focuses on building benchmarks for intrinsic dimension (ID) estimation. Pointing on that usual benchmarks for ID estimation are simple, the authors build a set of more complex physics-inspired manifolds. Building off of some tractable, well-understood manifolds (e.g., Stiefel manifolds, Grassmanians), the authors apply different embeddings of these classes into higher-dimensional spaces and put these forward as an ID benchmark. A suite of ID methods is evaluated on these benchmarks.

**Strengths:**

- The work is well-motivated - it is true that current benchmarks focus on very trivial manifolds that have no relationship to real-world data (though I'm not sure those proposed here have much relationship either - see weaknesses).
- This paper is well-written and I enjoyed reading it. This may be strange to say in a review, but the authors' passion for the subject really shines through in the level of detail, connections, and examples they give throughout.
- I do think the experiments are executed nicely. The analysis is unique and interesting, comparing performance to properties like curvature and noise (in addition to ID).

For these reasons, I am opting for an accept score.

**Weaknesses:**

- It's not clear at all whether these benchmarks are representative of real-world ID estimation problems. The work briefly mentions how Grassmanians and Stiefel manifolds have been studied in ML, but presumably not in a context when their IDs were unknown and needed to be estimated.
- Although some interesting quantitative analysis is performed showing *when* estimators fail, no insight is provided into *why* these failure cases exist on a per-estimator basis. I don't think I found in the work any suggestions on how to improve ID estimators on the basis of this benchmarking work either. (The notion of generating different embeddings of real-world manifolds is mentioned in the conclusion, but it's not at all obvious what this would entail.)
- Most benchmark manifolds are constructed using the "Gilmore-Perelomov" coherent-state method. But there is no accessible explanation of this concept for a general ML audience. There is a highly technical description in Appendix D.1 which I cannot follow. All I can glean is that it is a piece of math used somewhere in the process of generating embeddings.
	- You need to provide more accessible details about this. Is it or does it provide a function of some sort? Surely you can black-box most of the technical details while providing us with a precise description of its purpose here.

Side note: the citation style violates the ICLR 2026 formatting instructions: https://github.com/ICLR/Master-Template/raw/master/iclr2026.zip

**Questions:**

- Can you give a less technical explanation of what you are doing with the coherent state method?

---

> ### Author Response · Authors · 2025-11-22
> **Response to Reviewer wuJs**
>
> We thank the reviewer for their comments. We address them below.
>
> > * It's not clear at all whether these benchmarks are representative of real-world ID estimation problems. The work briefly mentions how Grassmanians and Stiefel manifolds have been studied in ML, but presumably not in a context when their IDs were unknown and needed to be estimated.
>
> We agree with the Reviewer that generating controllable manifolds that have known ID and resemble real-world data is the eventual goal of this research direction. We make a size-able step along this path by identifying that such manifolds will likely have anisotropic (i.e., direction-dependent) curvature. Further distorting the curvature of either our manifolds or MNIST does not lead to a drop in IDE performance, but distorting the (initially isotropic) curvature of a sphere does, suggesting that our manifolds may be accurately simulating real-world data in this particular aspect of complexity.
>
> > * Although some interesting quantitative analysis is performed showing when estimators fail, no insight is provided into why these failure cases exist on a per-estimator basis.
>
> We agree with the Reviewer on the lack of robust explainability of our results. In the new version, we perform experiments on lPCA [see Section 6] and observe that, for manifolds with isotropic curvature, the explained variances have a sharp drop at the ID, and the drop becomes smoother once we distort the manifold. As such, the dimension of embedded and undistorted spheres is easily picked up by PCA, as opposed to the dimensions of the other manifolds.
> We also observe a weak correlation between anisotropy and IDE performance for the angle-based methods DANCo and ABID (Fig 16). This makes sense since the anisotropy is a deviation from the expected isotropic (hyper-spherical) distributions assumed for these methods to work well.
>
> >I don't think I found in the work any suggestions on how to improve ID estimators on the basis of this benchmarking work either. (The notion of generating different embeddings of real-world manifolds is mentioned in the conclusion, but it's not at all obvious what this would entail.)
>
> Having observed the effects of non-isotropic curvature, we believe that methods should incorporate the curvature tensor in their construction. We have added a comment in the discussion along these lines.
>
> > * Most benchmark manifolds are constructed using the "Gilmore-Perelomov" coherent-state method. But there is no accessible explanation of this concept for a general ML audience. There is a highly technical description in Appendix D.1 which I cannot follow. All I can glean is that it is a piece of math used somewhere in the process of generating embeddings.
> You need to provide more accessible details about this. Is it or does it provide a function of some sort? Surely you can black-box most of the technical details while providing us with a precise description of its purpose here.
>
> We add a non-technical description of our methods (see red text) in Section 3, Homogenous spaces, followed by an explicit calculation for 2-spheres in Appendix D.1.1. We would be glad if the reviewer could let us know if the addition is sufficiently clear for a non-technical audience.
>
> > * Side note: the citation style violates the ICLR 2026 formatting instructions: https://github.com/ICLR/Master-Template/raw/master/iclr2026.zip
>
> We thank the reviewer for bringing this to our attention. The citation style has now been modified to conform to ICLR 2026 formatting instructions.
>
>
> > Questions:
> > * Can you give a less technical explanation of what you are doing with the coherent state method?
>
> We have now updated our paper with this information, as described above.
>
>
>
> We hope our responses have improved your perception of our paper. We will appreciate it if you can increase your score in that case. If not, we are happy to discuss any further concern

---

> > ### Comment · Reviewer_wuJs · 2025-11-25
> >
> > Thanks for the comprehensive response, which answers most of my questions. I maintain a positive opinion of this work and opt to keep my score.
> >
> > > We add a non-technical description of our methods (see red text) in Section 3, Homogenous spaces, followed by an explicit calculation for 2-spheres in Appendix D.1.1. We would be glad if the reviewer could let us know if the addition is sufficiently clear for a non-technical audience.
> >
> > Yes, I do think these are helpful additions. But it would be helpful at the bottom of Section 3 to explicitly mention that the procedure you are describing is the coherent state method.

---

> > > ### Author Response · Authors · 2025-11-26
> > >
> > > We thank the Referee for pointing this out. We've consolidated the text at the end of section 3 and beginning of section 4 to be more clear about the method we're describing.

---

### Official Review · Reviewer_q2Vs · 2025-11-01

**Soundness:** 2
**Presentation:** 2
**Contribution:** 2
**Rating:** 4
**Confidence:** 4

**Summary:**

This paper introduces QuIIEst (Quantum-Inspired Intrinsic-Dimension Estimation benchmark), a benchmark designed to evaluate intrinsic dimension estimation (IDE) methods on synthetic manifolds with known intrinsic dimensions. The authors construct several manifold families, including Stiefel, Grassmann, Flag, and Pauli manifolds, by leveraging the structure of homogeneous spaces (quotients G/H). The benchmark allows controlled sampling and noise perturbation, and the paper compares multiple IDE methods such as PCA, MLE, TwoNN, DANCo, ABID, and CorrInt on these manifolds.

**Strengths:**

- The benchmark provides a consistent way to evaluate intrinsic dimension estimators on known manifolds.
- The experimental section is well-organized, comparing several established IDE methods under controlled settings and providing clear quantitative results.

**Weaknesses:**

- Fundamentally, the work is unrelated to quantum mechanics. All constructions are standard results from Lie group representation theory and homogeneous space geometry. The “quantum-inspired” terminology appears unnecessary, as the methodology is fully explainable within classical differential geometry. To me, the quantum story is merely a coverage.
- All manifolds used in the benchmark are homogeneous manifolds such as Stiefel, Grassmann, and Flag manifolds generated by Lie group actions. Non-homogeneous manifolds are not included. This limits the benchmark’s ability to assess IDE methods on more realistic and asymmetric data manifolds.
- The description of the quantum background, especially in Appendix D, is insufficient and lacks rigor. Several notations are either undefined or unclear.

Overall, the paper mainly reuses known results on homogeneous manifolds to generate a dataset for IDE evaluation. The datasets comprise existing matrix manifolds. While the benchmark may be useful in practice, its conceptual novelty is limited.

**Questions:**

see wk

---

> ### Author Response · Authors · 2025-11-22
> **Response to Reviewer q2Vs**
>
> We thank the reviewer for their comments. We address them below.
>
> > * Fundamentally, the work is unrelated to quantum mechanics. All constructions are standard results from Lie group representation theory and homogeneous space geometry. The “quantum-inspired” terminology appears unnecessary, as the methodology is fully explainable within classical differential geometry. To me, the quantum story is merely a coverage.
>
> While the coherent-state method has a history outside of pure mathematics, we agree with the Referee that excessive quantum jargon may distract from our main message. As such, we have toned this down. In particular, we have tentatively proposed to the AC, SAC and PCs to change our title to  “How Now Oblong Cow: Benchmark for Intrinsic Dimension Estimation Using Homogeneous Spaces”, thereby omitting the word “quantum” and using 'homogenous spaces' instead.
>
> > * All manifolds used in the benchmark are homogeneous manifolds such as Stiefel, Grassmann, and Flag manifolds generated by Lie group actions. Non-homogeneous manifolds are not included. This limits the benchmark’s ability to assess IDE methods on more realistic and asymmetric data manifolds.
>
> In addition to homogeneous spaces, we also test how IDE performance changes upon distortion of our manifolds under random diagonal general linear transformations in Section 5.3. We observe that performance is not significantly affected by such distortion, indicating that homogeneous spaces seem to be already challenging enough.
>
> Having said this, we agree with the Referee on the importance of having a diverse data set going forward. We have added a way to make double coset spaces, which are not just non-homogeneous but also not manifolds. We also perform experiments on fractal non-manifolds in the Appendix. We highlight these contributions in the main text through added discussions.
>
>
> > * The description of the quantum background, especially in Appendix D, is insufficient and lacks rigor. Several notations are either undefined or unclear.
>
> We thank the Referee for pointing this out, as it has improved our manuscript. We have now made our derivations clearer and added a descriptive example to explicitly go through the steps in our recipe.
>
> > * Overall, the paper mainly reuses known results on homogeneous manifolds to generate a dataset for IDE evaluation. The datasets comprise existing matrix manifolds. While the benchmark may be useful in practice, its conceptual novelty is limited.
>
> We agree with the Reviewer that the mathematical results presented in the paper are known. We apply these results in a novel way to create a new benchmark for intrinsic dimension estimation (IDE). This benchmark highlights how traditional benchmarks with simpler “spherical-cow” manifolds fail to provide reasonable evaluation standards for real-world data. We highlight the similarities between real-world data and QuIIEst benchmark through additional experiments.
>
> > * Questions:
> see wk
>
> We hope our responses have improved your perception of our paper and will appreciate it if you can increase your score in that case. If not, we are happy to discuss any further concerns.

---

> ### Comment · Reviewer_q2Vs · 2025-11-26
>
> Thanks for the response.
>
> As I mentioned, the original version had little real connection to quantum mechanics and was essentially based on homogeneous spaces. The current manuscript has been revised accordingly. As the whole story and motivation has been changed, it is almost a new paper. I believe an additional round of careful review would still be appropriate.
>
> I, therefore, keep my original score.

---

> ### Author Response · Authors · 2025-11-26
>
> We thank the Referee for making this point, and concede that we should have better differentiated between what is new and what was already included in the initial submission. Our two primary changes were (1) the addition of MNIST experiments, and (2) a conjecture about non-isotropic curvature being the reason why IDE methods fail for our manifolds.
>
> Comparisons against previous benchmarks, as well as the anisotropic distortion, noise perturbation, and scaling behavior experiments—the “meat” of our calculations—were all included in the initial submission.
>
> The renaming of the paper was done to tone down references to the relevance of quantum mechanics to the construction of our dataset. This and other changes do not alter the key messages of our work.
>
> We hope that our response convinces the referee to upgrade their score.

---

> > ### Comment · Reviewer_q2Vs · 2025-11-28
> >
> > Thanks for the additional clarifications.
> >
> > I still have some remaining concerns regarding the shift in underlying motivation and mechanisms. Nevertheless, my evaluation is borderline, and I am open to either acceptance or rejection. If the other reviewers and the AC feel that no additional review round is needed, I am comfortable proceeding accordingly.

---

### Author Response · Authors · 2025-11-22
**Global Response**

We would like to thank the reviewers for their effort in providing constructive feedback, which made us  think more deeply about the project. We have now added several discussions in the updated manuscript to address the concerns raised. The new text is added in red to make it easier for reviewers to track changes. We are happy to discuss further to address any remaining concerns.

***Summary of reviews:***

***Strengths***: The reviewers generally agreed that the benchmark was novel (5gjM) in the context of Intrinsic Dimension Estimation (IDE) evaluation. They appreciated that QuIIEst provides a consistent, controlled way of evaluating IDE (q2Vs) and well-organized experiments (q2Vs, wuJs). They also acknowledged that the idea behind QuIIEst was well-motivated (wuJs). They also mentioned that QuIIEst can be a toy dataset for future projects involving manifolds (5gjM).

***Weaknesses***: Reviewer q2Vs mentioned that the major weaknesses of the work were no relation to quantum mechanics, no use of non-homogenous manifolds, lack of rigor in the technical description. Reviewer wuJs expressed concerns about relation of real-world data to QuIIEst manifolds, qualitative analysis of failure modes for different estimators and non-technical description of coherent state method. Reviewer 5gjM mentioned that our analyses on geometric properties should be better motivated and explanation of the estimators’ performance.

***Summary of major changes:***
1. We have tentatively modified the title, abstract and introduction to highlight our contributions.
2. We add experiments on IDE performance on MNIST. In particular, we perform the distortion experiments discussed in Section 5.3 (squeezing) and Section 5.4 (additive noise) on MNIST and demonstrate qualitatively similar results with analogous experiments on QuIIEst. This corroborates our claims that QuIIEst represents real-world data more closely.
3. We now add a heuristic explanation for why these estimators fail on QuIIEst. We conjecture that this is because Euclidean distances lead to biased estimators of local neighborhoods on manifolds with anisotropic curvature.
4. We add experiments examining why lPCA fails for QuIIEst.
5. We highlight our results on non-manifolds like fractals, and add discussions on non-homogenous spaces with double coset spaces.

---

### Author Response · Authors · 2025-12-03
**Final Thoughts**

We would like to thank the reviewers for their constructive feedback and productive discussions, which spurred us to think deeply about our problem, come up with reasonable conjectures explaining our results, and improve our manuscript.

In this project, we evaluate several intrinsic-dimension estimators (IDEs) on a family of homogenous spaces that we collectively call QuIIEst. We observe that these estimators fail on QuIIEst, with worse performance compared to prevalent benchmarks. We conjecture that this is because of the anisotropic curvature of QuIIEst manifolds. To corroborate our claim, we anisotropically distort manifolds further, which leads to no significant change in IDE performance. In contrast, manifolds with isotropic curvature show larger degradation in IDE performance. We show that IDE performance on MNIST similarly does not change upon distortion of that dataset, suggesting that QuIIEst strikes a balance between being sufficiently well-controlled and representative of real-world data.

**Strengths:** Reviewers generally agreed that the benchmark offers a consistent framework for evaluating ID estimators (q2Vs, wuJs, 5gJM). They highlighted that the experiments were well-executed (q2Vs, wuJs). They noted that the strong motivation for moving beyond trivial benchmarks (wuJs) and that providing large families of controllable manifolds, aka “toy datasets”, is useful for the community (5gJM).

**Weaknesses:**
Reviewers pointed out the following weaknesses:

(1) The “quantum-inspired’’ framing was unnecessary or insufficiently explained (q2Vs,5gJM).

(2) Failure mode analysis of different estimators was limited (wuJs) and it was not clear how these failures could be better understood or mitigated (5gJM).

(3) The benchmark includes only homogeneous manifolds, which may not be representative of real-world data (q2Vs, wuJs).

(4) Reviewer 5gJM also raised concerns about scalability and reproducibility.

(5) Our explanation of the coherent-state method was unclear (q2Vs, wuJs, 5gJM).

**Post-rebuttal views:** Our rebuttal addressed all of these concerns. Reviewer wuJs agreed to maintain their positive opinion of our work. Reviewer q2Vs maintained some doubt about the underlying motivation, but emphasized that their opinion was borderline. Reviewer 5gJM, who initially had a positive opinion, did not comment further before the new rules were instated.

**Our core contribution continues to be (1) to highlight inadequacies with current standards for IDE evaluation and (2) to provide a rigorous and representative benchmark.** We believe that controlled synthetic datasets are useful tools for the ML community in the context of IDE and manifold-learning problems, and ICLR is clearly an ideal platform to present this work. If accepted, we will release the code publicly with proper documentation for easy access.

---

### Meta-Review · Area_Chair_EmZT · 2025-12-20

**Summary:**

The paper is about creating new synthetic datasets of known intrinsic dimensionality that have non-trivial geometric properties which can be used for benchmarking intrinsic dimension estimators. The authors conclude that anisotropies are critical for mimicking real-world data, and claim that their benchmark resembles real world data more closely than previous benchmarks because they now include anisotropies. These conclusions are questionable.

Main concerns from reviewers that informed my decision:
- The work positions itself as quantum inspired, but this is unnecessary.
- All synthetic datasets come from quotients of Lie groups, meaning they are homogeneous, and not representative of real world data.
- When estimators fail on the constructed benchmarks, there is no indication of why they failed and how to fix them.

Given that reviewers were split, and the authors heavily updated their paper after reviews, I also read the paper in detail. I had additional concerns in the original and added content which I will cover below.

**Reviewer Concerns:**

- Quantum inspired (partially addressed):

I strongly agree with the critique that the paper unnecessarily positions itself as related to quantum physics. The authors changed their title, and modified parts of the paper to "downplay" this connection, however it is still strongly evident throughout the work. In reality, the method could be concisely described in terms of group theory and linear algebra only. I recommend that the authors do this, and largely remove the connection to quantum physics. As it stands, the Gilmore-Perelomov connection seems like it is added only to make the method sound sophisticated. Other physics terminology such as "spin squeezing" is irrelevant (and not explained in the text) when the actual mathematical description of the method is straightforward. The appendix's review on Quantum and ML begins with completely unrelated information about quantum computers. Physics notation is used in App D.1 without being defined and would likely not be familiar to ICLR readers

- Homogeneity, and real world data (partially addressed):

The authors mention adding double coset spaces in an appendix which are non-homogeneous (however this was not reviewed initially). They also argue in the rebuttal that homogeneous manifolds are already challenging enough. However this misses the point of the question. The objective should not be to find abstract mathematical objects where estimators fail, but to accurately represent properties of real-world data.

- Failure reasons (partially addressed):

This question is only addressed in the rebuttal for a single estimator, lPCA, with new content added to the paper after the rebuttal. However, from Fig 2, lPCA appears to be the single worst method - it may be more relevant to discuss failure modes of better estimators.

In addition, from my reading of the paper I had several major concerns with the original and added content.

- ID after perturbation

Throughout the experiments the text claims that ID estimators have increased error when some perturbations are made to the original manifold, but this assumes that the ID of the perturbed manifold matches the original manifold, which is probably an incorrect assumption. In any case, the authors do not justify this assumption. Hence, the conclusions the authors make based on these results are likely to be misleading.

- Claim of novelty around anisotropy

At several points the authors claim that their proposed method more closely represents real-world data compared to previous benchmarks due to anisotropy, but it is far from clear that this is true. Recent benchmarks listed in the authors' own literature review [A, B, C] all use anisotropic datasets of known dimensionality to benchmark. Specifically, Paper [A] uses multiscale manifolds; Paper [B] uses synthetic image manifolds; Paper [C] uses highly anisotropic data manifolds generated through random affine transformations or diffeomorphisms which both preserve ID. This latter benchmark already shows that the most performant pre-existing ID estimators have degraded performance on these anisotropic manifolds compared to the initial isotropic versions. Hence, some of the claims of novelty made by the authors may be overstated compared to work they were already aware of.


[A] Tempczyk et al. "LIDL: Local Intrinsic Dimension Estimation Using Approximate Likelihood" ICML 2022 (Note I viewed the published version - the authors' paper links to an outdated workshop preprint")

[B] Stanczuk et al. "Diffusion Models Encode the Intrinsic Dimension of Data Manifolds" ICML 2024 (Note I viewed the published version - the authors' paper links to an outdated preprint with a different title: "Your diffusion model secretly knows the dimension of the data manifold")

[C] Kamkari et al. "A Geometric View of Data Complexity: Efficient Local Intrinsic Dimension Estimation with Diffusion Models" NeurIPS 2024 (Note I viewed the published version - the authors' paper links to an outdated workshop preprint)

- Omission of model-based estimators, and some non-model based

This work does not benchmark against any model-based ID estimators, even though the authors cite several such papers including [A, B, C]. Model-based estimators are shown by these works to handle complex synthetic datasets better than the methods used by the authors. The ID estimators used were mostly from a standard open-source package (scikit-dimension) which actually contains quite a few additional estimators that were left out. In particular, work the authors cite [A, C] found the ESS estimator to be much stronger than MLE, so its exclusion is strange.

- The added content contains conjectures that are unsurprising or hardly supported. While conjectures are by their definition not proven, these conjectures may mislead readers more than they benefit.

Sec 5.6 now conjectures that real data is anisotropic, but this statement is not surprising at all. I have never seen an ML researcher suggest that real world data would lie on an isotropic manifold.

Sec 6 now claims anisotropic curvature is the most important factor in the failure of ID estimators, but this is stated after investigating only two other scalar curvature quantities which are not nearly enough to characterize curvature.

Sec 6 also now suggests that information from the Riemann curvature tensor should be added to ID estimators, but there is no indication that these quantities are feasible or practical to compute from data samples alone, nor is there any direct evidence in the paper that these quantities are related to ID estimation. Hence, the added material appears to be pure speculation which was not seen by the original reviewers.

- Citations

The updated paper misuses citation formats (\citet \citep). The authors made no effort to cite published versions of works rather than preprints, which can mislead readers trying to compare with existing work.


I am recommending to reject this work as it is overall unpolished, significant new material was added after initial reviews, and for the reasons given above. To improve the work I recommend the authors:
- Remove the connections to quantum physics, since they were not explained well and are ultimately unnecessary to the description of this method. The paper would be much more clear without physics jargon and instead relying solely on clear mathematical terms.
- Use the most accurate non-model based estimators, and include model-based estimators.
- Ensure claims that reference novelty compared to prior work are true and fair.
- Reduce the amount of conjecture and rely on firm analysis or robust experimentation to make conclusions.

**Reviewer Scores:**

Reviewer q2Vs - 4 -> 4 (As stated in a comment)

Reviewer wuJs - 6 -> 6 (As stated in a comment)

Reviewer 5gjM - 6 -> 6 (My estimation)

---

### Decision · Program_Chairs · 2026-01-26

Reject